

# Simulating model uncertainty of subgrid-scale processes by sampling model errors at convective scales

Michiel Van Ginderachter[1], Daan Degrauwe[1,2], Stéphane Vannitsem[1], and Piet Termonia[1,2]

[1]Royal Meteorological Institute, Brussels, Belgium
[2]Department of Physics and Astronomy, Ghent university, Ghent, Belgium

**Correspondence:** Michiel Van Ginderachter (michiel.vanginderachter@meteo.be)

**Abstract.** Ideally, perturbation schemes in ensemble forecasts should be based on the statistical properties of the model errors. Often, however, the statistical properties of these model errors are unknown. In practice, the perturbations are pragmatically modelled and tuned to maximize the skill of the ensemble forecast.

In this paper a general methodology is developed to diagnose the model error, linked to a specific physical process, based on a

comparison between a target and a reference model. Here, the reference model is a configuration of the ALADIN (Aire Limitée Adaptation Dynamique Développement International) model with a parameterization of deep convection. This configuration is also run with the deep convection parameterization scheme switched off, degrading the forecast skill. The model error is then defined as the difference of the energy and mass fluxes between the reference model with scale-aware deep convection parameterization and the target model without deep convection parameterization.

In the second part of the paper, the diagnosed model-error characteristics are used to stochastically perturb the fluxes of the target model by sampling the model errors from a training period in such a way that the distribution and the vertical and multivariate correlation within a grid column are preserved. By perturbing the fluxes it is guaranteed that that the total mass, heat and momentum remain conserved.

The tests, performed over the period 11 – 20 April 2009, show that the ensemble system with the stochastic flux perturbations

combined with the initial condition perturbations, not only outperforms the target ensemble, where deep convection is not parameterized, but for many variables it even performs better than the reference ensemble (with scale-aware deep convection scheme). The introduction of the stochastic flux perturbations reduces the small-scale erroneous spread while increasing the overall spread leading to a more skillful ensemble. The impact is largest in the upper troposphere with substantial improvements compared to other state-of-the-art stochastic perturbation schemes. At lower levels the improvements are smaller or neutral,

except for temperature where the forecast skill is degraded.



# 1  Introduction

Numerical weather prediction (NWP) can be described as solving an initial value problem with numerical models based on equations that describe the evolution of the atmospheric state and its interaction with other Earth system components. The solution of these equations is highly sensitive to the initial conditions (e.g. Thompson, 1957; Lorenz, 1969; Vannitsem and

Nicolis, 1997; Buizza and Leutbecher, 2015). As a result, the errors made in the initial conditions will lead to forecast error growth and eventually to a loss of predictability.

Quantification of the forecast uncertainty due to the uncertainty in the initial conditions is often done through ensemble-based Monte-Carlo simulations. It has been shown that such Monte-Carlo simulations, representing initial uncertainties consistent with the true distribution of initial condition errors, are typically under-dispersive and therefore lack reliability (e.g. Wilks,

2005; Palmer et al., 2005). This has motivated the research for methodologies to represent the uncertainties related to the model description in addition to the uncertainty in initial conditions.

The many simplifications and approximations that are necessarily made when constructing a numerical model from the laws of physics governing the evolution of the atmosphere, also contribute to the forecast error evolution (Lorenz, 1982; Dalcher and Kalnay, 1987; Orrel, 2002; Vannitsem and Toth, 2002; Nicolis, 2003, 2004; Nicolis et al., 2009). These model imperfections

arising from incomplete knowledge of physical processes, finite resolution, uncertain parameters in parameterizations and discretization, will also lead to a reduction in predictability even if the initial state would correspond exactly to the true state of the atmosphere. This category of errors, including filtering operations mapping the true atmospheric state on the state space of the model, is defined as the *model error* (Leutbecher et al., 2017). Recently, much effort is put in the development of schemes that simulate the random component of the errors of the model tendencies.

A first operational development of such a representation of model uncertainty was done by Buizza et al. (1999). Their scheme, generally referred to as Stochastically Perturbed Parameterization Tendency (SPPT) scheme, makes the assumption that the dominant part of the parameterized physics tendency error is proportional to the net physics tendency. Major revisions of the scheme were done by Palmer et al. (2009) and Shutts et al. (2011) both changing aspects of the probability distribution sampled by the SPPT scheme.

Another type of model uncertainty is addressed by Shutts (2005) and Berner et al. (2009). In their work, the focus lies on the model uncertainty associated with scale interactions that are present in the real atmosphere but absent in the model due to its finite resolution. To represent this uncertainty, they developed the Stochastic Kinetic Energy Backscatter (SKEB) scheme which introduces a stochastic streamfunction forcing determined by a local estimate of kinetic energy sources at the subgrid scale together with an evolving 3-dimensional pattern.

A different approach to account for model uncertainty is to perturb a set of key parameters within the parameterization schemes themselves. This technique was first applied in Bowler et al. (2008) and further adapted for use in a convection-permitting ensemble by Baker et al. (2014). The concept of stochastically perturbing parameters was further generalized in the Stochastically Perturbed Parameterization (SPP) scheme (Ollinaho et al., 2013, 2017). SPP extends the concept of perturbing parameters to perturbing locally both parameters and variables inside the parameterizations.



Keeping in mind the purpose of stochastic perturbations, which is to represent forecast uncertainties originating from model errors, the perturbations ideally have the same statistics as the error source. Since the model error sources are diverse and their statistical characteristics only partially known (Boisserie et al., 2014), the methods described above resort to pragmatic solutions: the amplitude of the perturbations or their spatial patterns are usually chosen such that a satisfactory reduction of ensemble under-dispersion is obtained (Berner et al., 2017).

Recent methods to assess sources of model uncertainty and their statistical properties are based on a comparison between perfect and target forecasts (Nicolis, 2003, 2004; Nicolis et al., 2009). This approach has been adopted in Seiffert et al. (2006) by comparing high and low resolution global circulation model (GCM) runs, or in Shutts and Pallarès (2014) by comparing the temperature tendencies associated with convection parameterizations in high and low resolution forecasts of the European Centre for Medium-Range Weather Forecasts (ECMWF) Integrated Forecast System (IFS).

The aim of the present paper is to use this method to characterize the model error related to a specific physical process by comparing two forecasts who differ only in the representation of the physical process under investigation. First a general description of the methodology is given, after which it is applied to deep convection. This is done by running the ALADIN (Aire Limitée Adaptation Dynamique Développement International) limited area model (LAM) (Termonia et al., 2018) in a configuration where deep convection is not parameterized and comparing it to a reference configuration with a scale-aware deep convection parameterization scheme. The simulations are performed over the tropics during a period of enhanced convective activity on a grid with a horizontal grid spacing of 4 km. This resolution lies on the verge of what is considered the convection-permitting scale, where it has been shown (Deng and Stauffer, 2006; Lean et al., 2008; Roberts and Lean, 2008) that non-parameterized convection often shows unrealistically strong updrafts and density currents as well as overestimated precipitation rates.

The model error is expressed in the form of a flux difference and its characteristics are obtained from a well-chosen training period. After discussing the statistical properties of the model error, its usefulness for probabilistic forecasting is investigated by developing a prototype stochastic flux perturbation scheme. The scheme introduces flux perturbations by sampling the model error from the training period. The impact of perturbing the fluxes, using the properties of the model error, on a short-range (48 h) LAM forecast is then studied, revealing a positive impact on the forecast quality.

The structure of this paper is as follows. The methodology for quantifying the model error and its statistical properties are presented in Section 2. In Section 3 the design of the perturbation scheme is described and the results of its application are discussed. Finally, the main results are summarized and a conclusions are drawn in Section 4.

## 2 Quantification of the model error

### 2.1 Methodology for a high-order NWP model

In the theoretical work on generic dynamical low-order models (e.g. Nicolis, 2003, 2004; Nicolis et al., 2009) the behaviour of the model error is investigated by comparing the evolution laws of an approximating model with the exact evolution laws. For these low-order problems, both the model and exact evolution laws can be written in an analytical form. Consequently, also the





evolution of the model error, given by the difference between the model evolution and the exact evolution, can be expressed formally. However, when dealing with the comparison of the evolution law of an NWP model and the reality, the evolution law for the model error cannot be formally expressed as the underlying dynamics of the reality is not fully known. Therefore in this paper, the model error will be estimated by comparing the model under investigation or target model with a reference model. Although the reference model is still deficient at some level, it is considered to exhibit a more correct representation of the atmospheric evolution.

In this paper such a target model vs reference model comparison is performed to quantify the model error related to a one specific physical process. In order to correctly quantify this error, it needs to be isolated, i.e. other reasons why the evolution of the reference model differs from the evolution of the target model must be eliminated. For that reason, the formulation of both models differs only in the representation of the physical process under consideration. Other parameters, such as the time step and vertical and horizontal resolution are taken identically. Finally, both models start from the same model state, and the error is evaluated over a single time step. The model error obtained this way, can be seen as a source of model uncertainty and it will be shown how its properties can serve as a guide when developing stochastic perturbation schemes. Undeniably, the representation of the physical process in the reference model will still have shortcomings compared to the true atmosphere. Therefore, the retrieved model error should rather be seen as a lower bound on the error made in the representation of the physical process.

The proposed methodology is generic and can be applied to all physical processes that can be represented by multiple schemes within the same model (e.g. radiation, turbulence, condensation, microphysics). In this paper, it is applied to the representation of deep convection in a model with a horizontal grid spacing of 4 km. The reference model uses a state-of-the-art scale-aware deep convective parameterization, developed specific for the convection-permitting transition regime (Gerard et al., 2009; Yano et al., 2018). The target (or approximate) model assumes the deep convection resolved. It's lack of deep convection parameterization will, however, lead to a significant error with respect to the reference model (Deng and Stauffer, 2006; Lean et al., 2008; Roberts and Lean, 2008).

All simulations are performed with the ALADIN NWP model, using the ALARO (ALadin-AROme) canonical configuration (Termonia et al., 2018), henceforth called the *ALARO model*. This LAM is developed focusing on a seamless transition from the meso-scale to the convection permitting scales (10 – 1 km) and runs operationally in 16 countries.

In this paper, the model is run in hydrostatic mode and uses a spectral dynamical core with a two time level semi-Lagrangian semi-implicit scheme. The vertical discretization uses a mass-based hybrid pressure terrain-following coordinate. An overview of the used parameterization schemes and a description of the scale-aware deep convection parameterization can be found in Appendix A.

When the ALARO model is run without deep convection parameterization, it is assumed that the turbulence (together with shallow convection) and resolved condensation schemes might compensate for the absence of parameterized convective transport, condensation and evaporation. The model error should thus be defined as the difference in total vertical transport,





condensation and evaporation flux:

$$\epsilon_\psi^{\text{trans}} = J_\psi'^{td} - (J_\psi^{td} + J_\psi^c) \qquad \text{with} \quad \psi = q_v, q_l, q_i, h, u, v \tag{1}$$

$$\epsilon_\varphi^{\text{cond}} = F_\varphi'^{st} - (F_\varphi^{st} + F_\varphi^c) \qquad \text{with} \quad \varphi = vl, vi, \tag{2}$$

$$\epsilon_\phi^{\text{evap}} = F_\phi' - (F_\phi^{st} + F_\phi^c) \qquad \text{with} \quad \phi = rv, sv. \tag{3}$$

where $J_\psi^{td}$ and $J_\psi^c$ represent respectively the turbulent diffusion and convective transport flux of water vapour ($q_v$), cloud water ($q_l$), cloud ice ($q_i$), enthalpy ($h$) and zonal ($u$) and meridional ($v$) momentum. $F_\varphi^{st}$ and $F_\varphi^c$ represent the stratiform and convective condensation flux from water vapour to cloud water ($vl$) and from water vapour to cloud ice ($vi$). Stratiform and convective evaporation fluxes from rain to water vapour ($rv$) and snow to water vapour ($sv$) are denoted by $F_\phi^{st}$ and $F_\phi^c$ respectively. Non-primed fluxes are those obtained when running with deep convection parameterization while primed fluxes

represent those using no deep convection parameterization.

As explained before, the model error is defined theoretically as the difference in the evolution (expressed as a time derivative) of the target and the reference model. In a numerical model, this time derivative is approximated by the evolution during a single time step. For the model error to be correctly quantified, it is important that both models start from the same atmospheric state. On the other hand, it should be acknowledged that the physics parameterization fluxes at the beginning of the forecast, the so-

called "spin-up period", typically are not representative, because the model is still trying to reach a balanced state. Therefore, the quantification of the model error is not trivial: when taking the difference after the first time step, the models are still spinning up, and the fluxes are not representative for the physics parameterizations effects, but when taking the difference at a later forecast time, both models runs will have diverged, and the difference is no longer a proper estimate of the model error. This issue is resolved by starting two runs with an active deep convection scheme from identical initial conditions.

These identical simulations are given 12 hours to spin up. After 12 hours the deep convection parameterization in one of the simulations is switched off and flux errors defined in Eqs. (1), (2) and (3) are diagnosed one time step later.

## 2.2   Construction of a model error database

The previous section explained how the model error due to the parameterization of deep convection can be quantified appropriately. This section describes how a database of model errors can be constructed. This database is not only useful to investigate

the statistics of the model error due to deep convection parameterization (Sect. 2.3), but it will also be the basis for a stochastic perturbation scheme that can be applied in an ensemble prediction system (Sect. 3).

In order to build a database with relevant weather cases (i.e. cases with active convection), the model simulations are performed over a tropical region including the Indian Ocean and Indonesia during a period of enhanced convective activity. The domain contains $1189 \times 349$ grid points and covers a region from $7°$ S to $5.5°$ N in the latitudinal direction and from $73.5°$ E

to $116°$ E in the longitudinal direction (Fig. 1). The model has 46 vertical levels and is run with a time step of 300 s.

The model runs are started every 3 hours between 00:00 UTC 2 April 2009 and 21:00 UTC 10 April 2009. This period is described in Waliser et al. (2012) as a period of enhanced convective activity due to the presence of three active equatorial waves. Initial and boundary conditions are provided using a double nesting technique where the ALARO model with a hori-



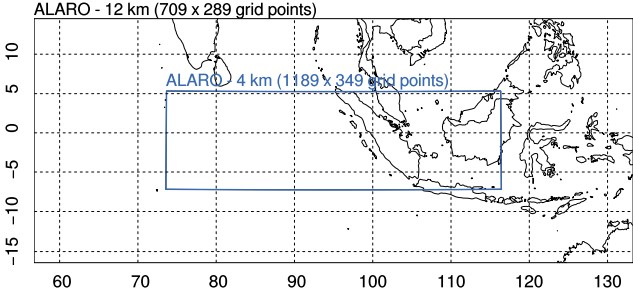

**Figure 1.** The ALARO 12 km grid spacing domain (outer black box) and the ALARO 4 km grid spacing domain (inner blue box).

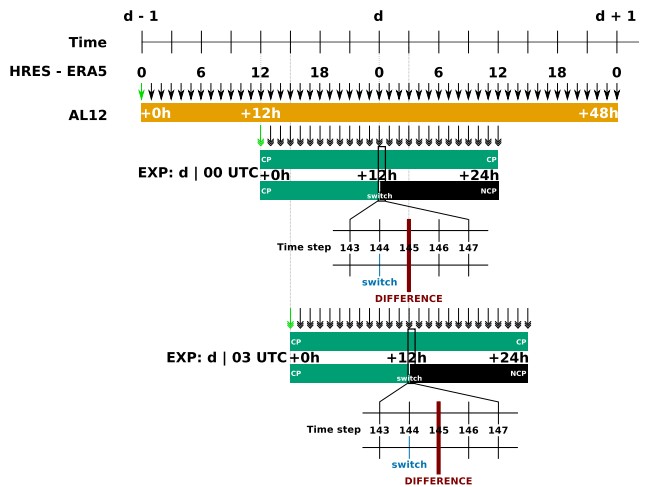

**Figure 2.** Organisation of the different runs used to evaluate the flux differences. Both runs start with deep convection parameterization (CP), while one of the forecasts switches off the parameterization after 12 hours (NCP). Here an example is shown for evaluation of the transport flux error at 00:00 UTC and 03:00 UTC. In practice such an evaluation is done every 3 hours between 00:00 UTC 2 April 2009 and 21:00 UTC 10 April 2009. A green single arrows indicates initial conditions were provided by the ERA5 reanalysis, while a black single arrow means that the LAM was forced with ERA5 boundary conditions. Double arrows indicate initial and boundary conditions are provided from the 12 km intermediate simulation.

zontal grid spacing of 12 km is driven by hourly ECMWF ERA 5 HRES analysis data (Copernicus Climate Change Service, 2017) and outputs hourly initial conditions (IC) and lateral boundary conditions (LBC) for the 4 km simulations. Organising the simulations this way leads to a total of 72 (9 days × 8 runs per day) evaluations of the model error on 8 different times in the day. A schematic overview of the organisation of simulations is given in Fig. 2.

5     A first assessment of the effect of deactivating the deep convection parameterization after 12 hours is done by comparing the domain-averaged water vapour transport flux and water vapour to cloud water condensation flux of a forecast where the deep convection scheme is deactivated after 12 h with the fluxes of a forecast where the deep convection scheme remains active (Fig. 3). During the first 12 hours both forecast configurations are identical. After the start of the forecast both the transport and





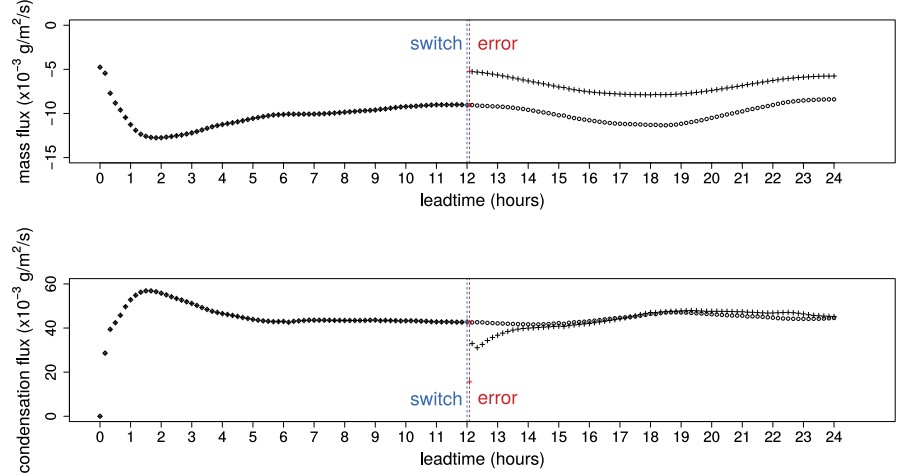

**Figure 3.** Domain averaged total transport fluxes ($J_v^{td} + J_v^c$ [open circles] and $J_v'^{td}$ [crosses]) for water vapour (top) and total condensation fluxes ($F_{vl}^{st} + F_{vl}^c$ and $F_{vl}'^{st}$) for cloud water (bottom). Both forecasts are started at 1200 UTC 5 April 2009 with deep convection parameterization, after 12 hours (blue dashed vertical line) the deep convection scheme is deactivated and both forecasts continue for another 12 hours. The differences between the fluxes are diagnosed after one time step (red dashed vertical line).

condensation fluxes intensify (negative transport flux indicates upward transport) during the first 3–4 hours, before reaching a balanced state. The chosen moment for deactivating the deep convection parameterization thus lies well outside of the spin-up regime.

The deactivation of the deep convection parameterization leads to an instant increase of the domain-averaged water vapour transport flux due to the absence of a (mainly) upward convective transport (Fig. 3a). The absence of an adaptation period indicates that the sudden removal of convective transport does not disturb the balance of the transport fluxes. Furthermore, the turbulent (including shallow convection) transport flux does not seem to compensate for the absence of a convective transport flux. Therefore, the total transport flux difference one time step after the switch can be considered as a representative measurement of the error in the transport flux as defined in Eq. (1).

The condensation flux (Fig. 3b), however, is no longer balanced to the new configuration and needs to readjust. This readjustment happens on two time-scales. A fast readjustment period during a couple of time steps is characterized by the sudden drop in total condensation flux. This reduction in the domain averaged condensation is caused by the re-evaporation of the convective cloud which is no longer protected by the deep convection scheme (see Gerard et al., 2009). On the time scale of a few of hours, the resolved condensation starts compensating for the absence of convective condensation. The evaluation of the difference in condensation and evaporation (not shown) fluxes as defined in Eq. (2) and (3) one time step after the deactivation of the deep convection parameterization is thus not representative for the typical error in condensation flux made when running without deep convection parameterization. Therefore, condensation and evaporation model error source remain problematic to diagnose since one finds himself either in the spin-up regime, when starting from the same initial conditions with different




configurations or in the readjustment regime when the convection parameterization is switched off after the spin-up period. On the other hand, comparing the fluxes when the model is readjusted to the new configuration, will no longer yield the isolated model error since the atmospheric state of both runs will have started to diverge.

Even though local differences of condensation fluxes can be large between both configurations, Fig. 3b shows that the
forecast without deep convection parameterization is able to restore the general condensation regime in the absence of the convective condensation. There is no similar compensation for the vertical transport, making the absence of convective vertical transport the dominant source of model error (considering the whole domain). Therefore, the remainder of this work will focus only on the error in the transport flux.

## 2.3   Properties of the vertical transport model error

Now that it is verified that above described methodology leads to a representative quantification of the transport error, the most important properties of the error are discussed below. To this goal, the error in transport flux of water vapour, cloud water, cloud ice, enthalpy and zonal and meridional wind is calculated for every grid point at every model level for all 72 reference - target simulation couples between 0000 UTC 2 April 2009 and 2100 UTC 10 April 2009.

### 2.3.1   Vertical profile

Figure 4 shows the average vertical profile of the transport flux errors of water vapour, enthalpy and zonal wind. The error in the water vapour transport flux (Fig. 4a) is positive (fluxes are counted negative upwards) at all levels but originates from different sources depending on the model level. The maximum error between 650–550 hPa corresponds to the underestimation of the upwards transport of moist air when there is no deep convection parameterization. The second smaller peak around 950 hPa, is associated with the missing downdraft transporting drier air to the lower regions and results in an underestimation of the
drying of the lowest levels when compared to simulations with deep convection parameterization. Differences in the transport fluxes of the cloud condensates (not shown) display similar characteristics.

Compared to water vapour, the error in enthalpy transport (Fig. 4b) peaks at a higher level (250–350 hPa). This peak is linked to the updraft parameterization, typically entraining warm air at its start and detraining it at higher levels. The reason the maximum transport flux error of water vapour lies at lower levels than that of enthalpy is explained by the condensation. At
higher levels condensation removes some of the water vapour from the updraft reducing its total transport. The smaller peak around 950 hPa in Fig. 4b coincides with that of Fig. 4a and is thus also linked to the downdraft transporting colder air down.

The interpretation of the transport flux error of zonal momentum (Fig. 4c) is somewhat counter-intuitive. Typically one would expect the error to be negative since the missing parameterized updraft transports air with lower momentum to higher levels where detrainment reduces the total momentum. The wind direction, however, is dominantly westward resulting in a
missing upward transport of slower, negative momentum and thus a positive error. At lower levels the downdraft transports high momentum eastward (positive) wind downward resulting in a (small) negative momentum flux error at 900–1000 hPa.

The standard deviations, shown in the bottom row of Figure 4, are typically one order of magnitude larger than the mean

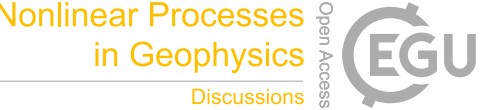

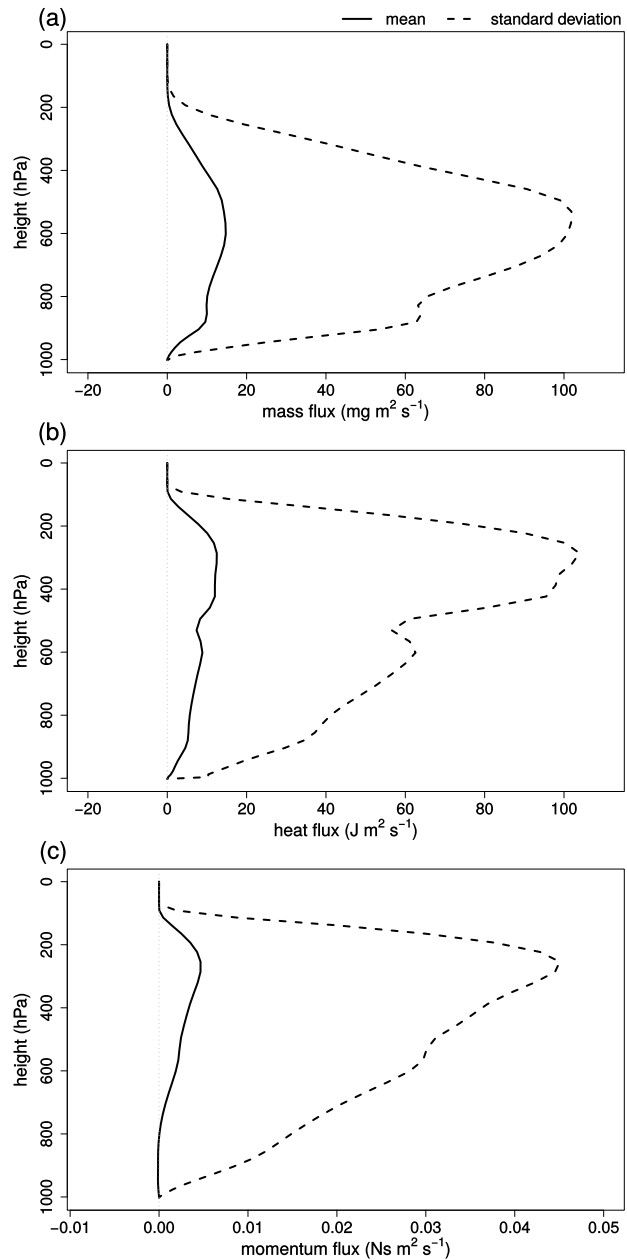

**Figure 4.** Vertical profile of the mean transport flux error (full line) and standard deviation (dashed line) for specific humidity (a), enthalpy (b) and zonal momentum (c).

of the respective errors. The convective transport is thus, besides being responsible for a systematic forcing, also an important source of variability for the tendencies in the middle and upper troposphere.




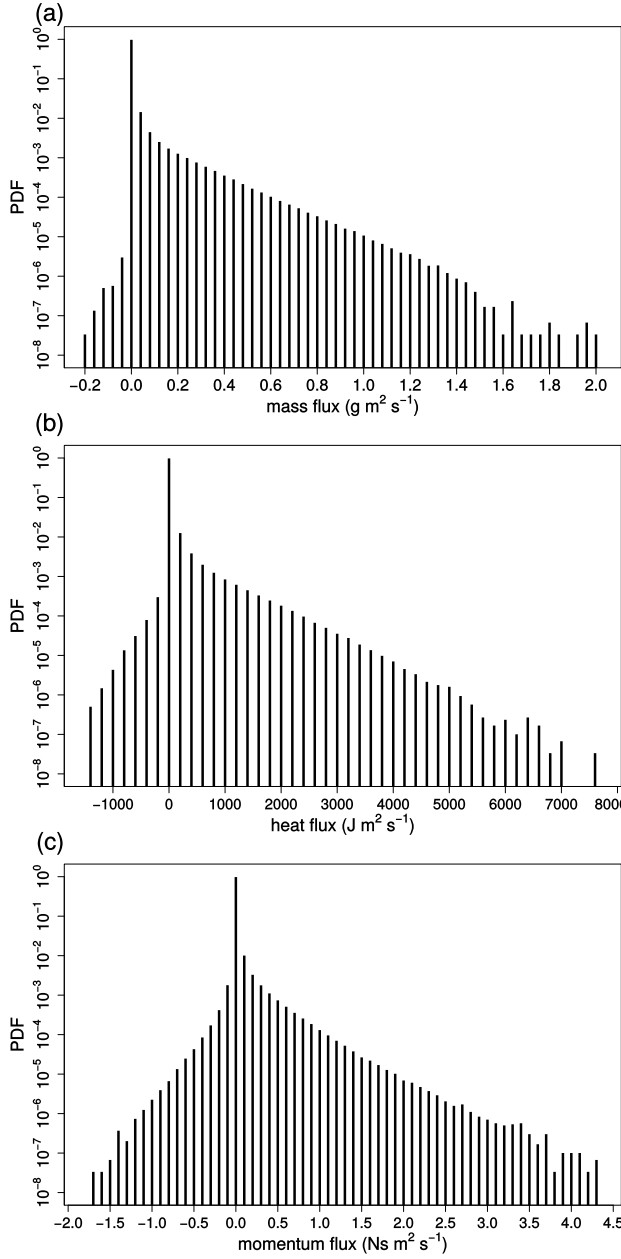

**Figure 5.** Probability density function (PDF) of the transport flux errors for water vapour (a), enthalpy (b) and zonal wind (c) at model level 20 ($\sim$ 250 hPa). In blue the fitted exponential distribution.



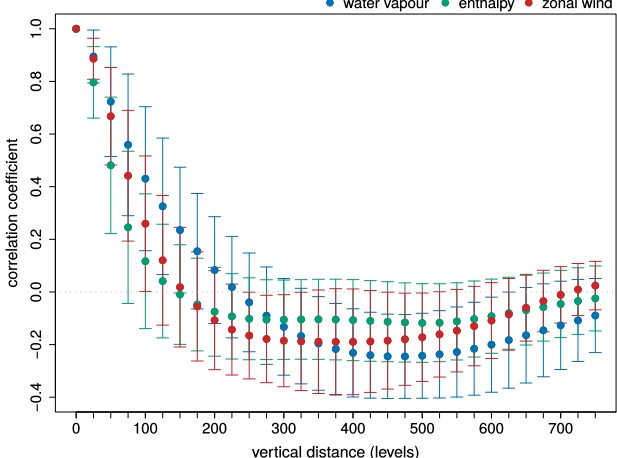

**Figure 6.** The vertical autocorrelation coefficients as a function of the vertical distance (in hPa) of the transport flux error for water vapour (blue), enthalpy (green) and zonal wind (red). The error bars show 1 standard deviation.

### 2.3.2 Probability distribution

Figure 5 shows the probability density distribution of the flux errors at the level of highest variability ($\sim 250$ hPa) for water vapour, enthalpy and zonal momentum. All three distributions show a large peak around zero, indicating that in a majority of the grid points no error is present. This peak represents the grid points without any convective activity. The long tail of the

5 distributions explains the large variance seen in Fig. 4. More specifically, the linear trend for positive values, most distinct for water vapour and enthalpy, hints at an exponential distribution of the transport error in grid points with convective activity. The number of grid points having a negative transport flux error is very small (0.01 % and 1.8 % of the total amount of grid points with non-zero error for respectively water vapour and enthalpy).

### 2.3.3 Vertical correlation

10 The vertical autocorrelation of the water vapour, enthalpy and zonal wind transport flux errors is shown in Fig 6. The correlation with vertically neighbouring grid cells is high for all three flux errors. The vertical correlation decays fastest for enthalpy, where the correlation drops below 0.2 after a distance of 75 hPa. Vertical correlation becomes negative for all errors for distances larger than 225 hPa. The vertical correlations indicate a clear vertical structure of the transport fluxes. Accounting correctly for this vertical structure is will be important when developing a stochastic perturbation scheme in Sect 3.

### 15 2.3.4 Inter-variable correlation

The transport flux errors of the different variables all originate from the absence of an up- and downdraft. Therefore, their inter-variable correlation is expected to be large. However, the correlation matrix (Table 1) shows only moderate correlation between water vapour, cloud water and zonal wind, while the correlation coefficient of the flux error of meridional wind w.r.t.





**Table 1.** Correlation matrix at model level 27 of the transport flux errors of water vapour ($q_v$), cloud water ($q_l$), enthalpy ($h$) and zonal ($u$) and meridional ($v$) wind.

|  | $\epsilon_{q_v}$ | $\epsilon_{q_l}$ | $\epsilon_h$ | $\epsilon_u$ | $\epsilon_v$ |
|---|---|---|---|---|---|
| $\epsilon_{q_v}$ | 1.0 | 0.55 | 0.49 | 0.43 | -0.05 |
| $\epsilon_{q_l}$ | 0.55 | 1.0 | 0.18 | 0.22 | 0.03 |
| $\epsilon_h$ | 0.49 | 0.18 | 1.0 | 0.28 | -0.02 |
| $\epsilon_u$ | 0.43 | 0.22 | 0.28 | 1.0 | -0.02 |
| $\epsilon_v$ | -0.05 | 0.03 | -0.02 | -0.02 | 1.0 |

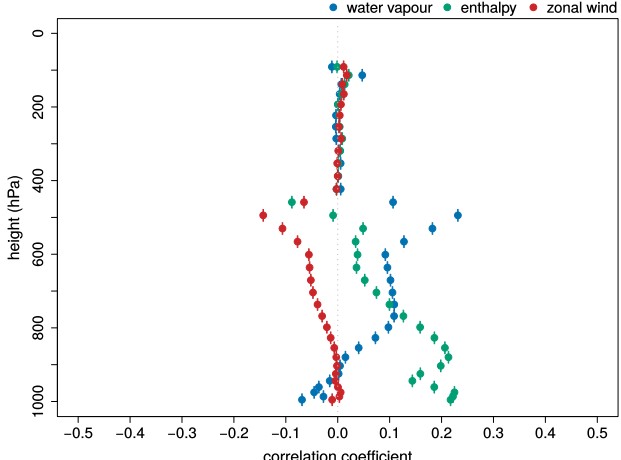

**Figure 7.** The Pearson correlation coefficients of the transport flux error w.r.t. the total transport flux of water vapour (blue), enthalpy (green) and zonal wind (red) at all model levels. The error bars show the 95 % confidence interval.

the other errors is close to zero. This is because, contrary to the other variables, meridional momentum flux error doesn't have a dominant sign. For the same reasons as discussed for the vertical correlation, an appropriate representation of these inter-variable correlations should be a major requirement when building a stochastic perturbation scheme.

### 2.3.5 Correlation with transport flux

5 Finally the correlation with the total transport flux of the target model is investigated. Large correlation coefficients between the transport flux and its error would suggest a linear relationship: $\epsilon_\psi = a J'^{\text{trans}}_\psi|_{\text{unpert}}$. In this case the perturbed fluxes ($J'^{\text{trans}}_\psi|_{\text{pert}}$) could be written in a multiplicative form: $J'^{\text{trans}}_\psi|_{\text{pert}} = (1 + a) J'^{\text{trans}}_\psi|_{\text{unpert}}$, as is done in the SPPT scheme. Figure 7 shows the correlation between the transport flux and its error for water vapour, enthalpy and zonal wind for all model levels. In general, the relation between a flux and its corresponding error is weak, with correlation coefficients ranging from -0.1 to 0.3. This





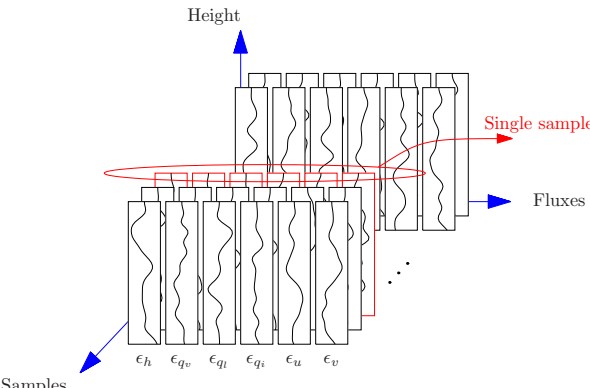

**Figure 8.** Schematic overview of the sampling of a grid column containing the different transport flux errors.

seems to suggest that a stochastic perturbation scheme based on a simple multiplication of the fluxes with a random number, will ignore part of the actual variability, resulting in an underdispersive ensemble system.

## 3   Random Perturbations for uncertainty forecasting

### 3.1   Perturbation scheme

When perturbing the transport fluxes during a forecast, the distribution of the perturbations ideally is the same as the distribution of the errors described in the previous section. Additionally, the perturbations should preserve the vertical, horizontal, temporal and inter-variable correlations of the errors. However, how to enforce all of these constraints simultaneously remains an open question, not only here but also in other schemes (Palmer et al., 2009). Here, a simple perturbation scheme is proposed that simulates appropriately the distributions (Fig. 5), while also preserving the vertical and inter-variable correlation of the

errors. As discussed above, the correct representation of both the vertical and inter-variable correlation of the transport flux perturbations is necessary to keep the perturbed fluxes physically consistent.

The vertical and inter-variable correlation are preserved by grouping the flux-error profiles per grid column and storing them in a database. During the forecast, the transport fluxes inside a particular grid column are perturbed by sampling a grid column from the database and subtracting the associated flux errors from the respective total fluxes (Fig. 8). The resulting perturbed

tendencies can thus be written as:

$$\frac{\partial \psi}{\partial t}|_{\text{pert}} = -g\frac{\partial}{\partial p}(J_\psi^{\text{tot}} - \epsilon_\psi^i), \tag{4}$$

with all flux-error profiles ($\psi = q_v, q_l, q_i, h, u, v$) belonging to the same sampled grid column $i$. Adding the error as a flux in a flux-conservative manner in Eq. (4) ensures that the total budgets of mass and energy are conserved by the perturbation.

The distribution shown in Fig. 5 contains all grid columns, including those without any convective activity, resulting in a



large number of cases where the flux error is zero. To prevent sampling these non-convective grid columns, they are excluded when building the database. Only grid columns where the convective activity is significant are retained by defining a cut-off updraft mass flux $M_{\text{cut-off}}$ and selecting only those grid columns which satisfy:

$$\overline{M_u} < M_{\text{cut-off}} \qquad \text{with} \qquad \overline{M_u}, \, M_{\text{cut-off}} < 0, \tag{5}$$

with $\overline{M_u}$ the updraft mass flux, available from the CP configuration, averaged over the vertical column. This allows us to retain only the error at those grid columns where convective activity is relevant. This is achieved by taking a value of $M_{\text{cut-off}} = $ -0.5 Pa s$^{-1}$.

   Finally, a proxy for deep convection is needed to determine the grid columns in the NCP run where the transport fluxes should be perturbed. Two different convection criteria are compared. The first criterion (labelled *MOCON*) uses the vertically

integrated moisture convergence in the PBL as a proxy for convective activity (Waldstreicher, 1989; Calas et al., 1998; van Zomeren and van Delden, 2007), while the second configuration (labelled *OMEGA*) uses the grid column averaged resolved vertical velocity. For both proxies a threshold value, $MOCON_c$ and $OMEGA_c$ respectively, is determined. During every time step of the integration, when the chosen threshold is exceeded for a given grid column, a sample is drawn from the database as shown in Fig. 8 and the transport fluxes are perturbed according to Eq. (4). Although the perturbations are sampled indepen-

dently in space and time from the database, the use of these thresholds introduces some crude spatio-temporal correlation in the flux perturbations.

   The choice of the threshold criteria $OMEGA_c$ and $MOCON_c$ is closely related to the one of the cut-off updraft mass flux $M_{\text{cut-off}}$. While $M_{\text{cut-off}}$ can be used to decide what part of the tail of the flux-difference distribution will be sampled from, $OMEGA_c$ and $MOCON_c$ are chosen such that the percentage of grid columns exceeding the threshold in the perturbed

forecast is roughly equal to the percentage of grid columns selected on the basis of $M_{\text{cut-off}}$. With $M_{\text{cut-off}} = -0.5$ Pa s$^{-1}$, $OMEGA_c$ was set to -0.2 Pa s$^{-1}$, while $MOCON_c$ was set to $4 \times 10^{-3}$ g m$^{-2}$ s$^{-1}$.

## 3.2   Results and Discussion

The above described perturbation scheme is tested during a 10 day period between 11 and 20 April 2009 over the same domain

as described in Sect. 2.2. Every day a 10-member 48-hour ensemble forecast is started at 0000 UTC. First only the perturbation scheme is used as a source of uncertainty. These forecasts are driven by the ERA5 HRES analysis. Second, the perturbation scheme is combined with IC and LBC perturbations provided by 10 members of the ERA5 EDA analysis. The scores of the atmospheric fields are calculated w.r.t. the ERA5 HRES analysis, while precipitation scores are calculated w.r.t. the TRMM satellite observations (Goddard Earth Sciences Data and Information Services Center, 2011). The scores used in this section to

evaluate the forecast are explained in detail in Appendix B.





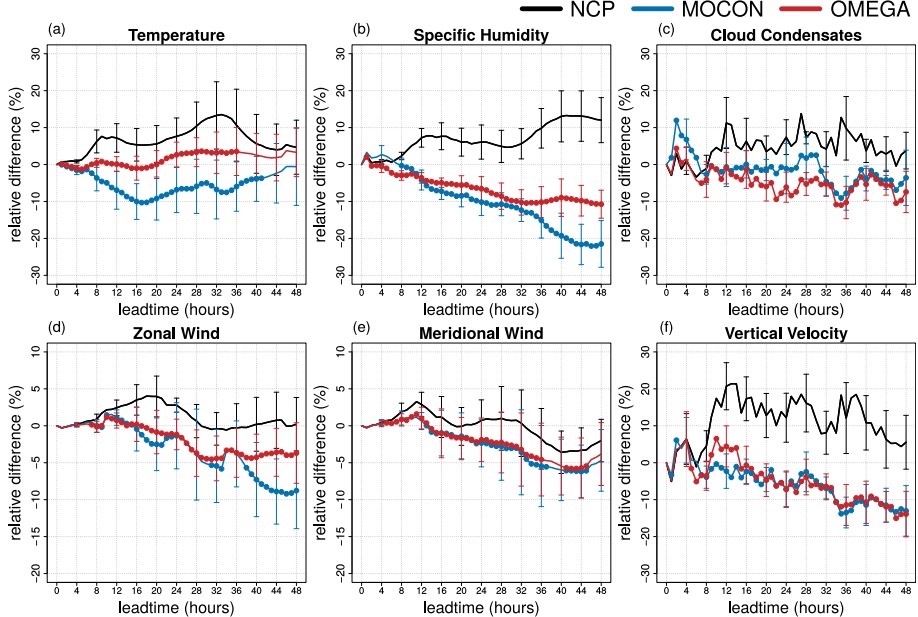

**Figure 9.** Relative RMSE difference (in % w.r.t. the CP configuration) of the NCP (black) control forecast (deterministic) and the MOCON (blue) and OMEGA (red) ensemble mean for temperature (a), specific humidity (b), cloud condensates (c), zonal (d) and meridional (e) wind and vertical velocity (f) at 250 hPa. Both ensemble forecasts are performed without perturbations in IC and LBC's. Error bars show the 95 % confidence interval. Leadtimes where the ensemble mean RMSE is significantly lower than the NCP control RMSE at the 95 % confidence level are indicated with a filled circle.

### 3.2.1 No IC and LCB perturbations

The direct impact of the stochastic forcing is studied by running both ensemble configurations MOCON and OMEGA without IC and LBC perturbations and comparing the RMSE of the ensemble mean (w.r.t. the ERA5 HRES analysis) and the RMSE of the deterministic NCP (target) forecast with that of the deterministic CP (reference) forecast.

5    The impact is largest in the upper atmosphere (Fig. 9). Here the RMSE of the ensemble mean of both ensemble configurations is significantly better than the NCP configuration for almost all variables and leadtimes. The RMSE of the MOCON configuration is even smaller than that of the CP configuration for temperature (significant during the first 24 h), specific humidity (significant for all leadtimes), zonal wind (significant during the last 12 h) and vertical velocity (significant after 24 h). The OMEGA configuration only performs better than the CP configuration for specific humidity (all leadtimes) and vertical

10   velocity (after 24 h).

This means that perturbing with flux errors indeed compensates the error made by disabling the deep convection parameterization, as was the immediate goal of these perturbations. But even better, part of error in the reference configuration due to the stochastic nature of deep convection, is correctly captured by the flux-error perturbation scheme.

In the lower atmosphere (Fig. 10) the effect of the stochastic perturbations is smaller. Here, the forcings have no significant





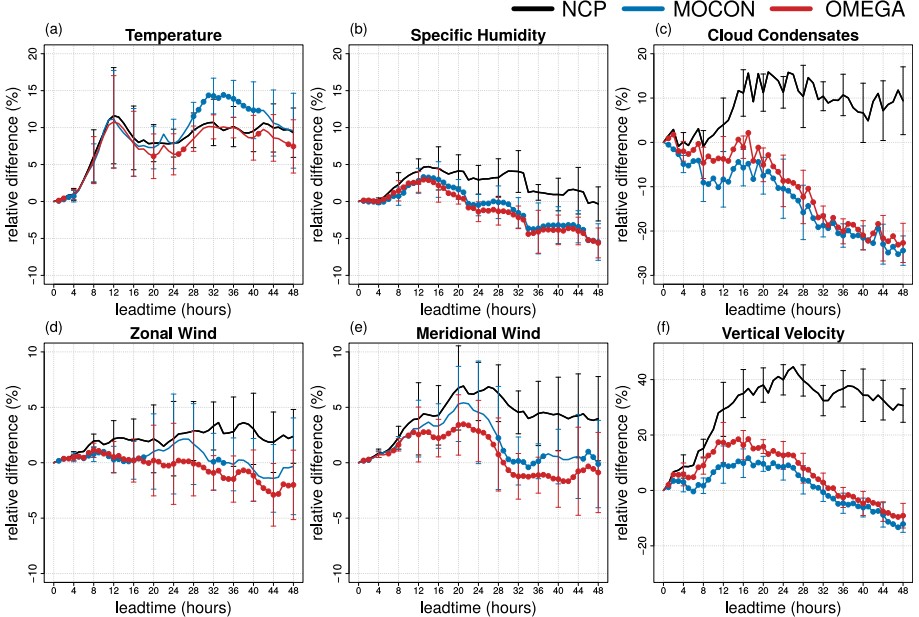

**Figure 10.** Same as Fig. 9 but for the 850 hPa level

impact on the temperature RMSE and also the reduction in specific humidity error is, albeit still significant, smaller. This is explained by looking back at profiles of flux differences in Figs. 4a and 4b. At 850 hPa the mean tendency perturbations caused by the addition of the corrective flux profiles is close to zero for both specific humidity and temperature, while also the standard variation of the corrective fluxes is smaller here than aloft. The situation for cloud condensates is opposite with a small but

significant improvement in RMSE at the 250 hPa level and a quite large improvement below at the 850 hPa level.

Figure 11 shows the spread created by the stochastic forcing from both perturbation configurations for specific humidity at 250 hPa at different leadtimes. The two ensemble configurations create similar spread both in amplitude and in spatial distribution, with largest spread concentrated in regions with convective activity. While the spread does show some spatial growth during the first 24 hours, the impact of the stochastic perturbations remains limited for regions with no convective activity.

Furthermore, the spread created during a convective episode does not seem to continue to grow once the convective episode is over and there is no longer a stochastic contribution to the tendencies. This is seen for instance over the Java Sea between Java and Borneo, where the spread induced after 24 hours has almost disappeared 24 hours later.

These results clearly shows that the proposed stochastic scheme provides a sensible way to generate perturbations. The effect of the stochastic perturbations is namely twofold. First of all they force the individual members closer to the CP forecast,

reducing the error. This is a logical consequence of perturbing the members by sampling the flux differences between target and reference forecast. On top of that, the spread is created at locations consistent with the error of the reference configuration, resulting in an ensemble that is more skillful than the reference CP forecast.



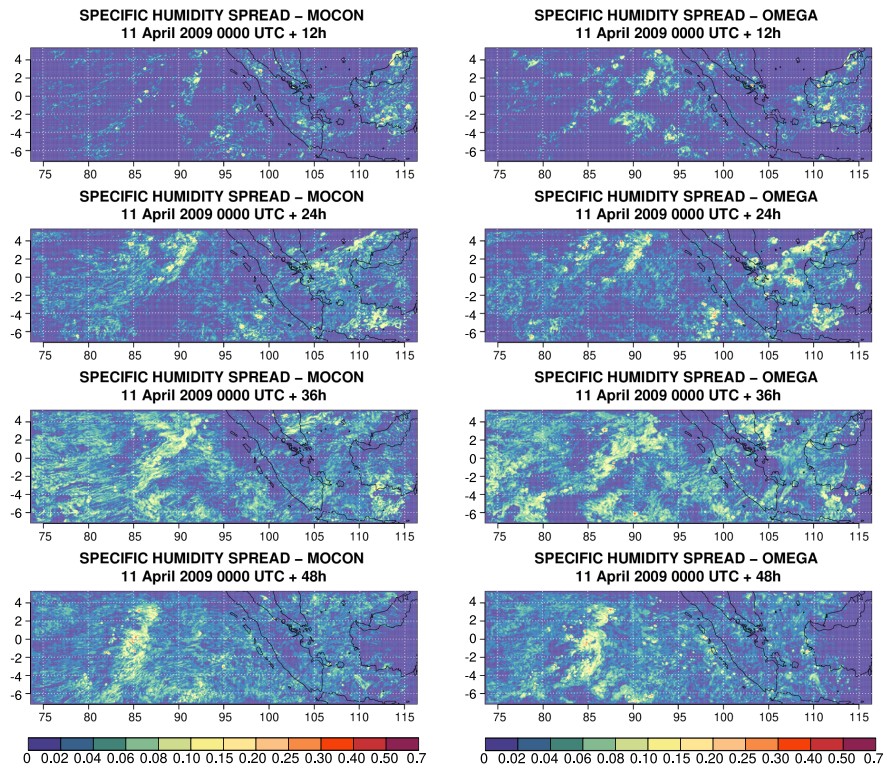

**Figure 11.** Snapshots of the specific humidity spread (g kg$^{-1}$) at 250 hPa for the MOCON (left) and OMEGA (right) ensemble without perturbations in IC and LBC's. Snapshots are shown for every 12 h during the 48-hour forecast started at 0000 UTC 11 April 2009

### 3.2.2 IC and LBC perturbations

Next, the interaction of the stochastic forcing with the IC and LBC perturbations is studied. The impact on the ensemble spread at 250 hPa is shown in Figure 12. In general the spread created by the stochastic forcing alone is much smaller (between 40 % and 60 %) than that created by the perturbed initial and boundary conditions. Only for cloud condensates and vertical
5   velocity, the spread induced by stochastic forcing alone approaches the spread generated by the perturbed initial and boundary conditions. The figure also shows that the combined effect of physics and IC/BC perturbations is smaller than the sum of the individual effects. This is in agreement with the findings of Baker et al. (2014) and Gebhardt et al. (2011).

For all variables the spread of the CP ensemble is smaller than that of the NCP ensemble. Comparing the snapshots of the zonal wind spread at 250 hPa of CP and NCP on 0000 UTC 13 April 2009 (+48h) (Fig. 13) shows a similar pattern of large
10   scale variability. The higher domain-averaged spread of the NCP ensemble is (dominantly) caused by increased small scale variability. This small scale variability is linked to regions with excessive updrafts resulting from the absence of a sub-grid stabilizing mechanism. It is however questionable that an increase in spread resulting from a inadequate representation of a certain process is an appropriate way to increase the skill of an ensemble.

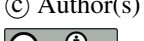



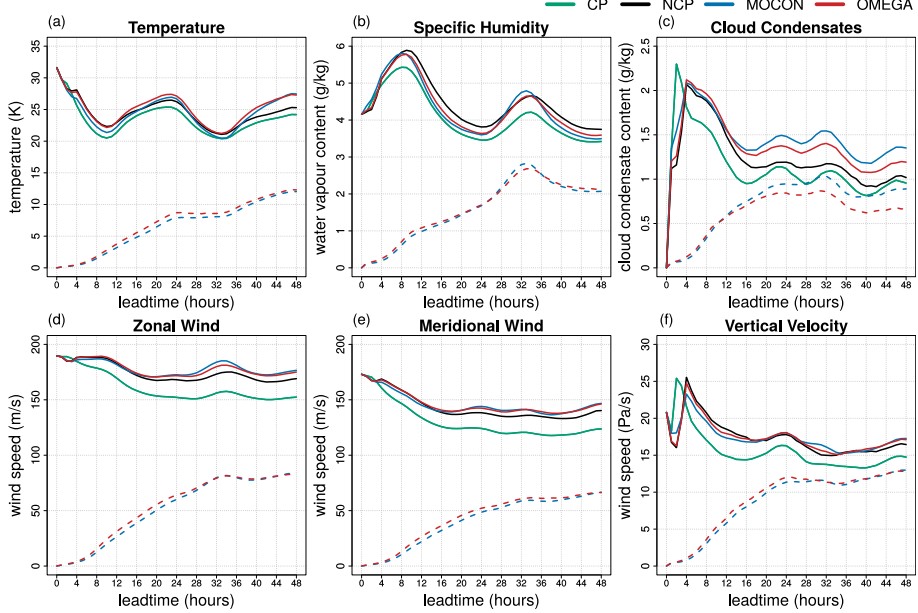

**Figure 12.** Ensemble spread of the MOCON (blue) and OMEGA (red) ensemble with (full line) and without IC and LBC perturbations (dashed line) together with the NCP (black) and CP (green) ensemble spread (created only by IC and LCB perturbations) at 250 hPa. Spread is calculated for temperature (a), specific humidity (b), cloud condensates (c), zonal (d) and meridional (e) wind and vertical velocity (f).

The addition of the stochastic perturbations increases the spread (between 4.9 % and 8.6 % w.r.t. the NCP ensemble) for all variables at 250 hPa except specific humidity. Fig. 13c shows that the increased spread is mainly caused by an expansion of regions with moderate spread (e.g. around 95° E), while regions of high spread now show a smoother pattern thanks to the introduced stabilizing fluxes. This is true especially for the specific humidity (not shown), where a decrease in spread is

combined with a large increase in skill (Fig. 14b). Here the stabilizing flux perturbations reduce the (erroneous) spread, while still maintaining enough small scale spread resulting in substantial increase in skill.

The impact of the flux perturbations on the probabilistic skill is analysed on the basis of the continuous ranked probability score (CRPS) calculated against the ERA5 HRES analysis. Figs. 14 and 15 show the relative difference in CRPS of the NCP, MOCON and OMEGA ensemble w.r.t. the CP ensemble for the upper and lower atmosphere respectively. The impact of the

stochastic forcing is highest in the upper atmosphere. As mentioned above, best results are seen for temperature and specific humidity, with significant improvements in specific humidity CRPS up to 24 % and 13 % for the MOCON and OMEGA ensemble respectively w.r.t. the CP reference ensemble and up to 31 % and 20 % w.r.t. the NCP target ensemble. The impact of the flux perturbations is less pronounced for the other variables, with smaller or non-significant improvements w.r.t. the NCP configuration.

Looking at the scores for horizontal wind, one can see that the CP reference ensemble performs worse than the target NCP ensemble after 22 (4) hours for the zonal (meridional) component. This is probably due to the smaller spread of the reference



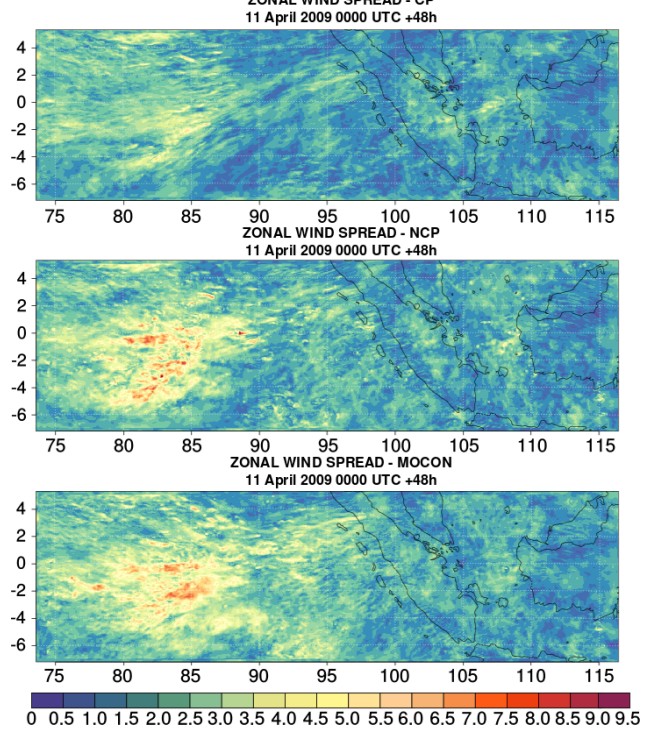

**Figure 13.** Snapshot of the zonal wind spread (m s$^{-1}$) at 250 hPa. Snapshot is taken at verification date 0000 UTC 13 April 2009 (leadtime 48 h) for CP (top), NCP (middle) and MOCON (bottom) ensemble, all with IC and LBC perturbations.

ensemble compared to that of the target ensemble (see Fig. 12) as explained above. Finally, for cloud condensates and vertical velocity the difference between target NCP and reference CP ensemble is small and also the impact of the stochastic forcings is limited, with the CRPS of the MOCON and OMEGA ensemble lying below the CP scores or between the NCP and CP ones, respectively. It seems that the spread created by the initial and boundary condition perturbations partly reduces the improved

5   skill of the CP configuration seen in Fig. 9.

At 850 hPa (Fig. 15), the most notable changes in CRPS scores are found in temperature, zonal wind and vertical velocity. Zonal wind scores of the MOCON ensemble have significantly improved w.r.t. both the NCP and CP ensembles, while the vertical velocity has improved w.r.t. the NCP ensemble only. Changes in 850 hPa temperature due to the stochastic forcing are up to 11 % and 5 % worse for the MOCON and OMEGA ensemble respectively. Differences in specific humidity and

10   meridional wind scores between NCP and CP ensemble are small and also the impact of stochastic forcings is mostly neutral.

For the upper troposphere zonal wind, the improvements (9 %) in skill are substantial when compared to other state-of-the-art stochastic perturbation schemes. In Ollinaho et al. (2017) improvements by the SPP and SPPT scheme around 2.5 % are reported for zonal wind at 200 hPa. Also for 850 hPa zonal wind CRPS improvements induced by the SPP and SPPT scheme (1.8 %) are comparable to the results presented here (1.7 %). The RP scheme presented in Baker et al. (2014) has a slightly



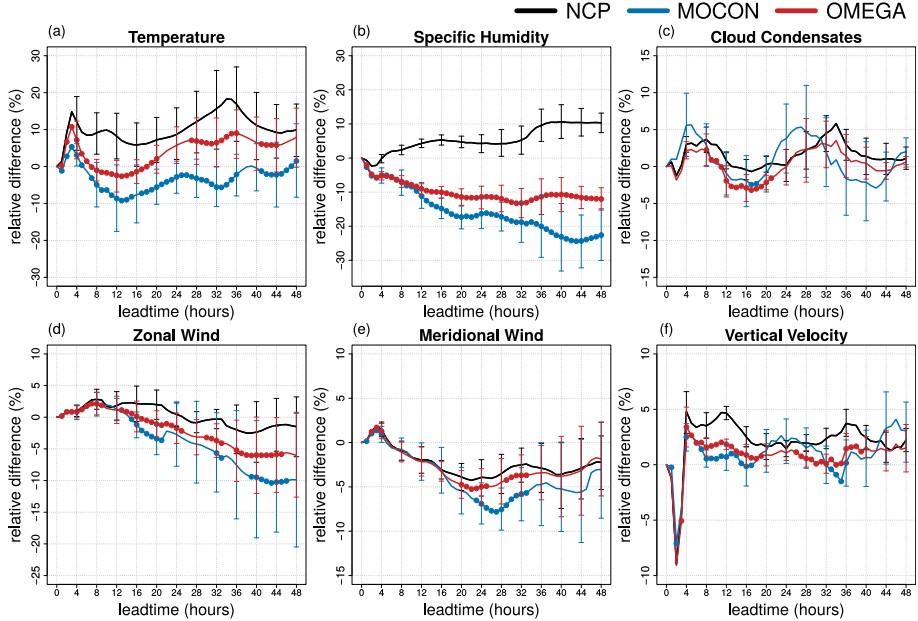

**Figure 14.** CRPS of the NCP (black), MOCON (blue) and OMEGA (red) ensemble for temperature (a), specific humidity (b), cloud condensates (c), zonal (d) and meridional (e) wind and vertical velocity (f) at 250 hPa. All ensemble forecasts are performed with perturbations in IC and LBC's. Error bars show the 95 % confidence interval. Leadtimes where the ensemble mean RMSE is significantly lower than the NCP control RMSE at the 95% confidence level are indicated with a filled circle.

larger impact with improvements around 15 % and 10 % for 1.5-m temperature and humidity respectively. However, one should keep in mind that only perturbations related to deep convection are considered here, while the SPP (RP) scheme perturbs 20 (16) parameters touching turbulent diffusion, convection, cloud processes and radiation.

Finally, the relative change in dispersion of the MOCON and OMEGA ensemble is investigated w.r.t. the NCP ensemble.
5   Table 2 shows the relative difference (as a percentage) in the number of times the ERA5 HRES reanalysis lies inside the range of the ensemble at leadtime 48 h. For all configurations the ensemble is underdispersive, a characteristic common to all single-model ensembles (Palmer et al., 2005). In the upper atmosphere (250 hPa) the flux perturbation reduce the under-dispersion for all variables. At this level the increase in number of observations falling within the ensemble is the result of a reduction in the bias of the individual ensemble members (not shown) and an increase in the spread. In the middle (500 hPa) and lower
10   (850 hPa) atmosphere, underdispersion is reduced for water vapour and zonal wind, while the stochastic ensembles are more underdispersive for temperature. The flux perturbations have little influence on the spread in the lower atmosphere (not shown). Therefore, changes in dispersion are mainly attributed to the positive (negative) shift in ensemble bias for specific humidity and zonal wind (temperature).





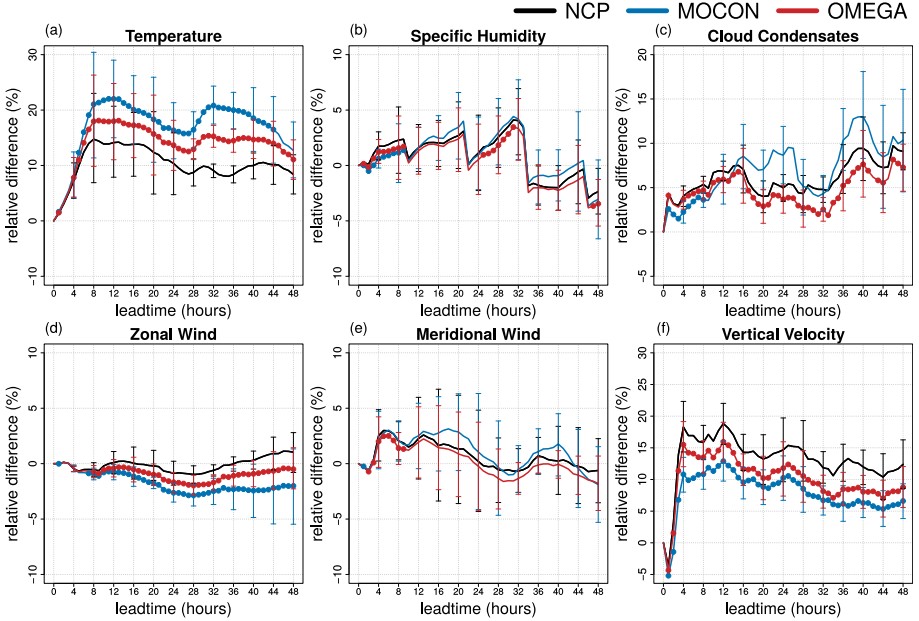

**Figure 15.** Same as Fig. 14 but for the 850 hPa level.

**Table 2.** Relative difference as percentage (w.r.t. the NCP configuration) in the number of times the ERA5 HRES reanalysis lies inside the range of the ensemble at leadtime 48 h for specific humidity, temperature and zonal wind. Values that are significant at the 95 % confidence interval are highlighted in bold.

|  |  | 250 hPa | 500 hPa | 850 hPa |
|---|---|---|---|---|
| Specific Humidity | MOCON | **19.2** | 2.2 | 1.6 |
|  | OMEGA | **15.3** | **4.0** | **0.9** |
| Temperature | MOCON | **16.9** | **-7.7** | **-6.5** |
|  | OMEGA | **11.1** | -2.0 | -2.0 |
| Zonal Wind | MOCON | **10.7** | **5.6** | 1.2 |
|  | OMEGA | **7.2** | **2.8** | 1.6 |

### 3.2.3 Precipitation

In what follows, the results for the OMEGA ensemble are very similar to those of the MOCON ensemble and are omitted for clarity. The ensemble skill for precipitation was evaluated using the CRPS calculated for the 3-hourly averaged precipitation w.r.t. the TRMM observations. The CRPS of all ensembles correlates well with the convective diurnal cycle (not shown). During a full day the 3-hourly averaged precipitation reaches a maximum at 0900 UTC (around 1600 local time) with a second smaller peak at 2100 UTC (0400 local time) (not shown). The peaks in CRPS at leadtimes 9 h, 21 h and 33 h are closely linked to these maxima of precipitation. Investigation of the domain-averaged precipitation amounts (not shown) reveals that the ab-




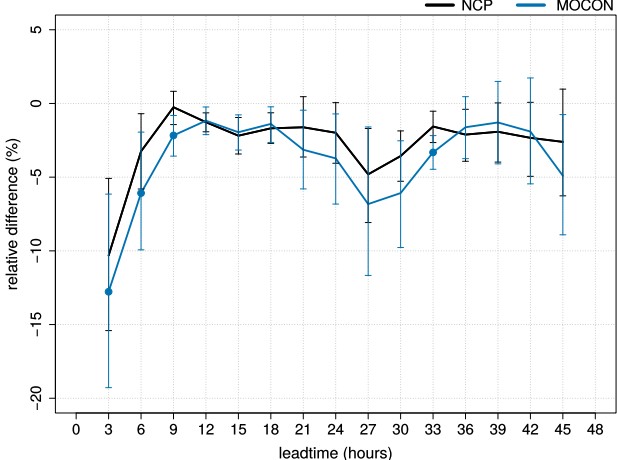

**Figure 16.** Relative change in CRPS w.r.t. the CP ensemble of 3-hourly averaged precipitation for the NCP (black) and MOCON (blue) ensemble. All ensembles include IC and LBC perturbations. Th error bars show the 95 % confidence interval. Leadtimes where the CRPS of MOCON is significantly different from that of NCP are indicated with filled circles. The CRPS is calculated w.r.t. TRMM observations.

solute error in domain-averaged precipitation is constant throughout the day, meaning that the diurnal pattern in the CRPS is rather caused by a mismatch of local precipitation regions than by an overall misrepresentation of the convective diurnal cycle.

Figure 16 shows the relative change in CRPS w.r.t. the CP ensemble. The impact of the absence of a deep convection parame-
terization and stochastic forcing on the 3-hourly averaged precipitation is relatively small. For all leadtimes, the NCP ensemble is more skillful than the CP ensemble despite the smaller RMSE of the individual CP ensemble members (not shown).

The improvement in skill of the MOCON ensemble is only significant during the first nine hours, afterwards the impact is neutral. This is an improvement compared to Baker et al. (2014) who only found improved skill in precipitation rate during the first two hours with their RP scheme and a worsening afterwards. A significant improvement in CRPS is also seen at lead time
33 h, coinciding with a maximum in convective activity. Unsurprisingly, the convective activity must be high in order for the stochastic forcing to have a significant effect on the precipitation.

In order to differentiate between different precipitation regimes, the Brier skill score (BSS) w.r.t. the CP ensemble for 24 h cumulated precipitation is shown in Fig.17 for different thresholds. No significant difference is seen between the NCP and MOCON ensemble at any threshold. Both ensembles do perform significantly better than the CP ensemble at the thresholds 1
and 5 mm day$^{-1}$. It seems that for low precipitation rates, the consistency between the different members of the CP configura- tion (a result of the deep convection parameterization), is penalized. The spread created by local grid point storms, on the other hand, is rewarded in the setting of an ensemble forecast. This result can be attributed to the same syndrome of better scores for the wrong reasons as discussed in Sect. 3.2.2. At larger thresholds no significant difference in BSS between the CP and both NCP and MOCON configurations is found. These results further confirm the suggestion that the (transport-) fluxes from the
deep convection scheme have a limited impact on the precipitation amounts.



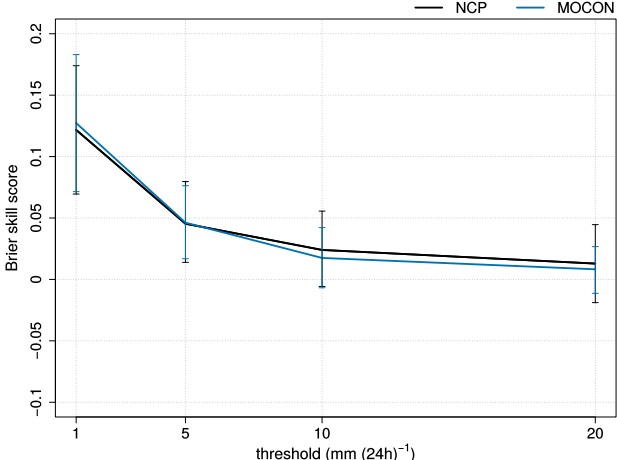

**Figure 17.** The Brier skill score w.r.t. the CP ensemble of the NCP (black) and MOCON (blue) ensemble for 24-hour accumulated precipitation at different thresholds. The Brier score is calculated w.r.t. TRMM observations. The error bars show the 95 % confidence interval.

## 3.3 Time to Solution

In conclusion, the computing-performance of the different configurations is studied. To this end, the time-to-solution of a single 48 h forecast was measured for all configurations. The reference CP configuration has the longest time to solution (2213.71 s; on 4 CPU's with 24 cores, Intel Xeon EP E5-2695 V4). The deep convection parameterization has a large impact on the run time. Deactivating it shortens the run time by almost 12 %. Both stochastic configurations have a small impact on the time-to-solution. With an increase in run time of 1.24 % w.r.t the NCP configuration, the MOCON configuration has the fastest time-to-solution (between CP, MOCON and OMEGA). Most (1.04 % of the total run time) of the extra run time is spent selecting the active grid columns and adding the fluxes. Loading the flux-difference database only occupies a small fraction of the extra run time (0.03 % of the total run time) but does consume around 30 % more memory. Relative to the NCP configuration the OMEGA forecast is 1.36 % slower. The added time w.r.t. the MOCON configuration stems from additional calculations needed to diagnose the column-averaged vertical velocity.

## 4 Conclusion and Outlook

This paper presents a novel approach for perturbing physics parameterizations of numerical weather prediction systems. The novelty lies in the idea that the perturbations are sampled from a database of model error estimates. In particular, the model errors studied here originate from switching off the parameterization of deep convection, thus degrading the forecast skill. The error is then diagnosed by taking the difference of the contributing physics fluxes of the runs with and without parameterization. Analysis of the error fluxes reveals the importance of the vertical correlation structure, as well as of the inter-variable correlation. Next, it is investigated whether these stochastic flux perturbations can increase the probabilistic forecast skill in an



ensemble context in such a way that it compensates for the lack of the parameterization.

The sampling method was implemented in a 10-member ensemble system testbed driven by the members of the ERA5 analysis. From the tests it follows that the ensemble system with the stochastic perturbations along with the IC and LBC perturbations not only compensates for the absence of the parameterization scheme, but for many variables it even outperforms

the ensemble system with the deep convection scheme switched on.

The ensemble without the deep convection parameterization exhibits too much spread as the result of an inadequate representation of the convective processes at the convection-permitting scales. The introduction of the stochastic perturbations has a stabilizing effect on the convective adjustment and reduces this small-scale erroneous spread while increasing the overall spread (except for specific humidity), leading to a more skillful ensemble. The impact is largest in the upper troposphere with

significant improvements in forecast skill for temperature, specific humidity and both zonal and meridional wind. At 850 hPa, improvements are smaller or neutral, except for temperature where the forecast skill is degraded.

An increasing number of EPSs are now entering the convection-permitting scales (4 - 1 km). Most of them are based on the assumption that deep convection is explicitly resolved (e.g. Gebhardt et al., 2008; Schwartz et al., 2010), even though the threshold resolution for turning off the deep convective parameterization is not well established. It may be advocated (see

e.g. Yano et al., 2018) that the transition towards fully convection-resolved resolutions should be carried out gradually. The stochastic perturbation approach presented here could be seen as such an intermediate approach avoiding the application of a scale-aware parameterization while still dealing with the ensuing model errors in a probabilistic sense.

The results shown here were produced in a controlled testbed, relying on perfect boundary conditions from the ERA5 reanalysis. To be able to optimally exploit the features of this approach within an operational EPS system it would be helpful to

address a few issues.

For instance, a better understanding of the strong moisture readjustment regime after disabling the deep convection parameterization within the error-diagnostics procedure could be useful. It could be investigated how to estimate the condensation flux differences by protecting the convective condensates from re-evaporation after disabling the deep convection scheme. If a controlled adjustment could be developed then the errors in the condensation and evaporation fluxes might also be taken into

account in the stochastic perturbation, potentially leading to more probabilistic forecast skill.

Combining the vertical correlation and multivariate correlation between the different flux perturbations with adequate spatio-temporal correlations would be useful, possibly relying on principle component analysis methods. In addition, the definition of the conditions determining the convective activity of a grid column and triggering the perturbations might be improved following the suggestions made in Banacos and Schultz (2005).

The creation of the sampling database requires extra implementation efforts and maintenance in an operational context. Additional research could be useful to model the error statistics and to find some universal error functions that may become candidates for PDFs to draw from for the stochastic sampling. The distributions found in Fig. 5 provide a good starting point for fitting such curves. If the functions are parameterized with a few parameters, the cumbersome computing task for estimating the error statistics may be replaced by a tuning exercise. Also the demanding task of building and maintaining the sampling

database would not be necessary.



In Subramanian and Palmer (2017) the work on stochastic tropical convection parameterization done by Khouider et al. (2010), Frenkel et al. (2012) and Deng et al. (2015) is used in an ensemble context and the probabilistic features of the so called ensemble superparameterization are compared to those of a classical SPPT approach. It would be useful to extend their comparison with the approach suggested here. The process-oriented perturbation method, studied in this work, is indicated

as highly desirable in Leutbecher et al. (2017) and is complementary to the different representations of model uncertainty presented in the introduction.

*Code availability.* The ALADIN Codes, along with all their related intellectual property rights, are owned by the Members of the ALADIN consortium and are shared with the Members of the HIRLAM consortium in the frame of a cooperation agreement. This agreement allows each Member of either consortium to license the shared ALADIN-HIRLAM codes to academic institutions of their home country for non-

commercial research. Access to the codes of the ALADIN System can be obtained by contacting one of the Member institutes or by sending a request to patricia.pottier@meteo.fr and will be subject to signing a standardized ALADIN-HIRLAM License agreement.

## Appendix A:  The ALARO model parameterization schemes

The ALARO model, used in this study, uses the "p-TKE"-scheme for the parameterization of the boundary layer mixing. This scheme complements the method described in Louis (1979) by a prognostic equation for the Total Kinetic Energy (TKE)

(Geleyn et al., 2006). Shallow convection is part of the turbulence scheme through a modification of the Richardson number (Ri) as described in Geleyn (1986). The ALARO model can be run with and without parameterized deep convection. In the case of parameterized convection, the scale-aware 3MT-scheme of Gerard et al. (2009) is used. This scheme calculates vertical transport fluxes as well as condensation fluxes which contribute to the gross cloud condensates passed to the microphysical scheme. The microphysical processes are parameterized similar to the scheme described in Lopez (2002), while a statistical

sedimentation formalism proposed by Geleyn et al. (2008) is used to calculate the precipitation and sedimentation fluxes.

The coupling between physics and dynamics is handled by an interface based on a flux-conservative formulation developed in Catry et al. (2007) and further generalized in Degrauwe et al. (2016). The interface transforms the fluxes coming from the parameterizations to tendencies using a clean description of the thermodynamic and continuity equations, ensuring the conservation of heat, mass and momentum.

## Appendix B:  Forecast skill

Evaluating the impact on the forecast skill of the stochastic forcing in the absence of perturbations in initial and boundary conditions is done on the basis of root mean square error (RMSE) and ensemble spread. Root mean square errors are calculated



with respect to ERA5 HRES reanalysis:

$$\mathrm{RMSE_{ctrl}} = \sqrt{\frac{1}{nx \times ny} \sum_{i,j=1}^{nx,ny} (x_{i,j}^{\mathrm{ctrl}} - x_{i,j}^{\mathrm{era}})^2}$$

$$\mathrm{RMSE_{stoch}} = \sqrt{\frac{1}{nx \times ny} \sum_{i,j=1}^{nx,ny} (\overline{x_{i,j}^{\mathrm{stoch}}} - x_{i,j}^{\mathrm{era}})^2},$$

with $x_{i,j}^{\mathrm{era}}$ and $x_{i,j}^{\mathrm{ctrl}}$ the respective reanalysis and control (CP and NCP) forecast values of field $x$ at grid point $(i,j)$ and $\overline{x_{i,j}^{\mathrm{stoch}}}$ the ensemble (MOCON and OMEGA) mean of field x at grid point $(i,j)$ and $nx$ and $ny$ the number of grid points in the $x$ and $y$ direction respectively.

The spread of the ensemble created by the stochastic forcing is given by:

$$\mathrm{spread}^{\mathrm{stoch}} = \frac{1}{nx \times ny} \sum_{i,j=1}^{nx,ny} \sigma_{x\,i,j}^{\mathrm{stoch}},$$

where

$$\sigma_{x\,i,j}^{\mathrm{stoch}} = \sqrt{\frac{1}{n-1} \sum_{k=1}^{n} (x_{i,j,k}^{\mathrm{stoch}} - \overline{x_{i,j}^{\mathrm{stoch}}})^2}.$$

The impact of introducing the stochastic forcing together with perturbed initial and boundary conditions on the skill is expressed using the CRPS (Brown, 1974; Matheson and Winkler, 1976). The CRPS in point $(i,j)$ measures the distance between the probabilistic forecast $\rho$ of variable $x$ and the reanalysis value $x_{\mathrm{era}}$ as

$$\mathrm{CRPS}_{i,j} = \int_{-\infty}^{+\infty} (P(x_{i,j}) - P_{\mathrm{era}}(x_{i,j}))^2\, dx$$

with $P(x)$ and $P_{\mathrm{era}}(x)$ cumulative distributions:

$$P(x) = \int_{-\infty}^{x} \rho(y) dy \qquad \text{and}$$

$$P_{\mathrm{era}}(x) = H(x - x_{\mathrm{era}}),$$

where

$$H(x) = \begin{cases} 0 & \text{for } x < 0 \\ 1 & \text{for } x \geq 0 \end{cases}$$

is the Heaviside function. The CRPS can also be interpreted as the integral of the Brier score (BS) over all possible thresholds of the considered variable.




The Brier skill score (BSS) used in Sect. 3.2.3 is defined as

$$BSS = 1 - \frac{BS_{ens}}{BS_{CP}},$$

with $BS_{ens}$ the Brier score of the ensemble under investigation and $BS_{CP}$ the Brier score of the CP ensemble. The Brier score (Wilks, 2011) defined as

5 $$BS = \frac{1}{nx \times ny} \sum_{i,j=1}^{nx,ny} (f_{i,j} - o_{i,j})^2,$$

where $f_{i,j}$ is the probability of a given event occurring in the ensemble forecast and $o_{i,j}$ a binary indicator of the occurrence of the event in the TRMM observations.

*Competing interests.* The authors declare that they have no conflict of interest.

*Acknowledgements.* The authors would like to thank Luc Gerard, Pieter De Meutter and Olivier Giot for useful discussions and suggestions.
10 All computations were performed and figures made with R (R Core Team, 2017) and the Rfa (Deckmyn, 2016) package. This work is supported by the Belgian Federal Science Policy Office under contracts BR/121/A2/STOCHCLIM.





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
