# Peer review of "Simulating model uncertainty of subgrid-scale processes by sampling model errors at convective scales"

_Nonlinear Processes in Geophysics, 2019_

## Referee Comment (RC1) · Anonymous Referee #1 · 11 Jun 2019

Review of "Simulating model uncertainty of subgrid-scale processes by sampling model errors at convective scales" by Ginderachter et al.

The authors present a methodology to diagnose model error associated with deep convection. They create a data-base of the diagnosed error and construct a stochastic parameterization of Monte-Carlo-type to sample the error to apply perturbations to their "target" models' total transport flux terms. While the paper is well written, and the construction of the database ensuring a vertical and cross-variable correlation provides some strength to the methodology, I do not recommend publication of this manuscript in its current form, as the study lacks justification of its basic theoretical assumptions,

mainly:

-The difference between the two model configurations is not the model error, but rather one representation of model uncertainty. At 4 km resolution, this uncertainty will pertain systematic differences between the two configurations chosen in this study, and sampling from the data base would mean consistently sample perturbations with the same systematic error. Using the differences between two configurations where one has a known systematic deficiency needs to be better justified, if at all possible.

-The main short-comings of this study is the computation of the "error" (uncertainty) itself. Here the authors turn off the deep convection parameterization and claim "it is assumed that the turbulence (together with shallow convection) and resolved condensation schemes might compensate for the absence of parameterized convective transport". And they proceed to compute the "error" as the difference in total transport (where one experiment is now missing the convective transport terms). This assumption is highly questionable. Just because there is no parameterized convection contributing to the transport flux of e.g. specific humidity, doesn't mean that there is no convective transport. In the "no parameterized convection" experiment this is now taken care of by the resolved dynamics, and the "compensation" discussed will be seen in the tendency of the dynamics. In fact, the authors do point out in the introduction that studies have shown that turning off the convective parameterization at $\sim$4 km can lead to unrealistically strong updrafts. What is the scientific justification for systematically adding a positive (or stronger negative) perturbation to the total transport when the convective transport is missing by construction, and is now resolved?

-The simulations should be made with non-hydrostatic dynamics, for the dynamics to be able to (have a chance) to realistically simulate vertical motions generated by convection.

-The perturbations are applied to the model considered the "target" forecast – which does not use a convective parameterization. Now you systematically introduce a larger

parameterized convective transport in a run with resolved convection. This seem to imply that the scale awareness of the model impose a reduction of the resolved convective transport, such that the improvement that you see relative to the control run (e.g. Figure 9), basically comes from again implicitly "activating" the convective parameterization (by systematically introducing a larger convective transport in the physics parameterizations).

-Why the first time-step after turning off the deep convection parameterization is a representative time of the model uncertainty needs to be better justified. The uncertainty due to convection ought to grow as a function of lead time. Figure 3 simply shows total transport with and without convection.

-Another aspect that is rather confusing in the experiment design is why the study is constructed such that the perturbations are applied to the model configuration that has no deep convection at 4 km, when the operational model uses the convective parameterization? Wouldn't it be more desirable to create a perturbation scheme that could be applied to the operational ensemble system at that resolution?

-Lastly, the perturbations in the distribution are not applicable to any general model system, but tied to this very particular experiment setup, and thus does not provide a general guidance for development of stochastic parameterizations. What happens if the model is used at 10 km or 1 km? Which configuration is now considered the 'perfect' model?

I can certainly appreciate the effort going into this work, and I hope the work on the cross-variable perturbation technique, which is a clear strength of this paper, can be published in some form. If the editorial decision of 'manuscript revision' is reached, I would suggest putting more emphasis on this aspect. I would also urge to change the underlying theory diagnosing the 'model uncertainty' rather than 'model error'. The reason I do not suggest major revision rather than reject is that I find the experimental design and underlying assumptions very unsatisfactory. One way to add some

scientific justification to the study would be to have the target model being the current operational configuration, then the 'perfect' model would be a very high resolution (non-hydrostatic) convection resolving version that you coarse grain back to the 4 km grid. Here you can compute the 'true' sub-grid variability of the flux, and use this in your data-base to perturb the transport fluxes in the 4km run deep convection scheme. The method you propose is just systematically correcting a "flawed" system, there is no random error component involved.

---

## Author Comment (AC1) · 17 Jun 2019

**Short reply to Referee nr. 1** Referee nr. 1 correctly raises the issue: "the study lacks justification of its basic theoretical assumption". But, in fact, the study is based on the theoretical analysis of Nicolis (2003, 2014) and Nicolis et al. (2009) (which are cited). The referee is nevertheless right that a clear description of the justification is lacking in the manuscript. Below we give one that was included in a previous version of the manuscript, but that we, unfortunately, removed to keep the paper concise. We now realize that this may be crucial to make the paper readable, since one can, rightly so, not expect the reader to check the equations in the cited papers, even if they are

correctly cited.

In fact we compute $\epsilon$ in Eq. (6) below in a full NWP model and this theoretically defines what we mean by model error. It also defines what the "reference" is, namely $X$. We have also proven that removing the deep convection degrades the forecast, see Fig. 1 (we will add this to the revised manuscript), so we believe this $\epsilon$ can be called a model error. By carefully reducing the experiment to a difference in the contributions of a few controllable terms we ensure that the theory of Nicolis (2003) can be applied. The corresponding terms in the NWP model are shown in Eqs, (1)-(3) in the submitted manuscript. The convenient thing about the deep convection parameterization is that the terms can be nicely isolated (and in fact allows for generating perturbations that conserve energy). One could take different terms, but for a first study we limit it to deep convection, as it is an essential ingredient of the atmospheric dynamics at scales smaller than 10 km.

That said, this would not be possible if we have different set ups for the dynamics, e.g. in the resolution, or by changing the hydrostatic to a non-hydrostatic version, since than we cannot reduce the model error to a few terms and would prevent us to effectively separate the model error from initial condition errors.

We very much appreciate the comment of Referee nr.1 and will give a more detailed point by point reply in the final response. We plan to include the paragraph below in the manuscript, either in the full text or in the appendix.

**Theoretical justification** For low-order models a characterization of the model error has been done by Nicolis (2003), who investigates the dynamical and probabilistic aspect of the model error in the absence of initial condition errors, and Nicolis et al. (2009) who study the evolution of prediction error under the combined effect of initial

condition and model errors. Essentially, the method used in Nicolis (2003) is described as follows: the respective correct evolution laws and model equations can be formally written as

$$\frac{d\mathbf{X}}{dt} = \mathbf{f}(\mathbf{X}) \tag{1}$$

$$\frac{d\mathbf{Y}}{dt} = \mathbf{g}(\mathbf{Y}). \tag{2}$$

Starting from identical initial conditions $\mathbf{X}(t_0) = \mathbf{Y}(t_0)$ one can write for a sufficiently small time interval $\delta t$:

$$\mathbf{X}(t_0 + \delta t) = \mathbf{X}(t_0) + \mathbf{f}(\mathbf{X}(t_0))\delta t \tag{3}$$

$$\mathbf{Y}(t_0 + \delta t) = \mathbf{X}(t_0) + \mathbf{g}(\mathbf{X}(t_0))\delta t. \tag{4}$$

The model error can be written as $\mathbf{U}(t) \equiv \mathbf{Y}(t) - \mathbf{X}(t)$ by extracting Eq. (3) from Eq. (4)

$$\mathbf{U}(t + \delta t) = (\mathbf{g}(\mathbf{X}(t_0)) - \mathbf{f}(\mathbf{X}(t_0))\, \delta t. \tag{5}$$

The model error source, determining the short time behaviour of $\mathbf{U}$ is then characterized by estimating

$$\epsilon = \mathbf{g}(\mathbf{X}) - \mathbf{f}(\mathbf{X}). \tag{6}$$

The aim of this paper is to study the feasibility of this method for estimating the model error source related to the absence of a parametrized convective forcing in a high order limited area model (LAM). This is done by running the ALADIN (Aire Limitée Adaptation Dynamique Développement International) numerical weather prediction (NWP) model in a configuration where deep convection is not parametrized and comparing it to a reference configuration with a scale-aware deep convection parametrization scheme.

**References**

Nicolis, C.: Dynamics of Model Error: Some Generic Features, J. Atmos. Sci., 60,
2208–2218, https://doi.org/10.1175/1520- 0469(2003)060<2208:DOMESG>2.0.CO;2, 2003.

Nicolis, C.: Dynamics of Model Error: The Role of Unresolved Scales Revisited, J. Atmos. Sci., 61,1740–1753,https://doi.org/10.1175/1520-0469(2004)061<1740:DOMETR>2.0.CO;2, 2004.

Nicolis, C., Perdigao, R. A. P., and Vannitsem, S.: Dynamics of Prediction Errors under the Combined Effect of Initial Condition and Model Errors, J. Atmos. Sci., 66, 766–778, https://doi.org/10.1175/2008JAS2781.1, 2009.

[Figure]

**Fig. 1.** RMSE (compared to the ERA 5 re-analysis) of the different model output variables at 850 hPa for the configuration with (green) and without (blue) convective parameterization.

---

## Short Comment (SC1) · 28 Jun 2019

In general, this is an interesting paper, which tackles the very hard "model-error problem" and has many high-quality elements.

**What I like in the manuscript?**

1. The careful examination of the spinup process when comparing tendencies from the two models. Without eliminating or radically reducing the initial transient there is no chance to detect the (very small in one time step) model error.

2. The flux formulation, which guarantees conservation of the spatial mean values of the perturbed fields. (I would note that the perturbations of the fluxes need to vanish at the boundaries to ensure the conservation.)

3. The careful way the perturbations from the archive are added to the model fluxes.

4. It's interesting that the error appears to be additive rather than multiplicative (section 2.3.5).

**What don't I understand or don't fully agree with in the manuscript?**

1. Actually, the authors study an artificially introduced model error. They do so by switching off the deep convection parameterization and examining the differences in the tendencies between the model-*with*-the-parameterization (the "truth") and the model-*without*-the-parameterization (the "target"). In other words, they study the error in the model-*without*-the-parameterization (with respect to the model-*with*-the-parameterization). In practice, however, we are interested in studying model errors/uncertainties *due to* a parameterization, not uncertainties due to the absence of a parameterization. Here I agree with Referee #1. Correspondingly, in practice, model-error perturbations are to be added to the (operational) model *with* the parameterization.

   However, I think this doesn't make the results presented in the manuscript irrelevant. They are still interesting – if regarded as an example of how an uncertainty in the model can look and how can it be modeled.

2. Essentially, the proposed technique is a substitute for a deep convection parameterization. The technique needs to be trained on a model with the parameterization or, as suggested by Referee #1, on a fine-grid model. The results (section 3) show that the substitute is quite successful. This is interesting by itself, but is

this approach practical? I think, it can be practical in the case when a physics pa-
rameterization is absent and the technique is trained on a high-resolution model.

3. The lack of spinup in Fig.3 (top) is remarkable. But it is shown for the *domain average* water vapor flux (so that there is no spinup in the *bias*). How about the **RMS** flux error: is it still small at the first steps after the deep convection parameterization is switched off? (I mean, switching off a parameterization should change the model attractor so that a transition of the model state from the old attractor to the new one is expected to take place, which should manifest itself as an adjustment process.)

4. The perturbation generation scheme is meaningful (as I mentioned above) but seems ad-hoc. In particular, it is unclear why different criteria are selected in sampling the archive and in perturbing the fluxes ($M_{\mathrm{cut-off}}$ versus OMEGA/MOCON). To justify the technique, you could require that the perturbations are sampled from the same conditional distribution of the error given the current state of the column in question.

**Specific comments**

1. Fig.4. Given the limited number of cases (72) and the very large standard deviations as compared to the mean values, the mean values seem to be not statistically significant. Even if we assume that the cases are independent and the distribution is Gaussian, the significance level can be easily seen to be well above the acceptable level of 0.05.

2. Figs. 3 and 4 display results for the *mean* fluxes. Figs. 5–7 and Table 1, however, seem to show results for individual profiles. I think it should be clarified which conclusions you draw from the mean-flux results and why the mean-flux results are relevant at all? I mean, in the perturbation generation scheme, you

seem to take individual profiles. But the distributions of individual profiles and the distributions of their domain averages should be very different.

3. I don't understand how the vertical and cross-variable correlations as well as the probability densities you have estimated are used in the perturbation generation scheme? Sampling profiles from an archive does not require any knowledge of their probability distributions. How do you actually use the distributions? From the manuscript, it seems that these are not used.

**Minor comments**

1. It is not everywhere clear at which level the fluxes are evaluated.

2. Fig.5. "In blue the fitted exponential distribution."

   I don't see anything blue in Fig.5.

3. P.11–12. "while the correlation coefficient of the flux error of meridional wind w.r.t. the other errors is close to zero. This is because, contrary to the other variables, meridional momentum flux error doesn't have a dominant sign."

   Actually, the correlation coefficient is insensitive to biases.

I am sure the paper will be improved and eventually published!

---

## Referee Comment (RC2) · Anonymous Referee #2 · 2 Aug 2019

The author use a hydrostatic model at 4km resolution without and without a cumulus parameterization (CP) as a basis to construct a probabilistic cumulus parameterization which amount to sampling the flux error distribution between the model with CP and the that without CP. While the procedure and the whole concept is very simple and may seem appealing the paper itself is very poorly written in the sense it is very confusing and hard to really understand that the take-home message really is. This is addition to many other flaws and unjustified choices made in the study. The use of a hydrostatic model at 4 km and expected to represent convective flows in the tropics realistically is, forgive the word, an aberration. The small improved seen with the use of the stochastic are arguably due to this shortcoming more than anything else. It is

well known that although too coarse to represent the details of individual clouds, CRM at few kilometre resolution (up to 10 km in some cases) represent well organized convection in the tropics; the Japanese did their first global CRM simulation at 7 km and got very realistic MJO, CCWs and MCSs. On the other hand it is also well known that the hydrostatic balances messes up gravity waves at scales of 50 km and less and a fortiori convective flows in this range. The way the sampling of the flux errors is done is not very clear. While I am likely confused by their narrative, the choose of the 250hP level as a reference for "sampling the grid column database" is not only not justified by the authors but it is also not accurate. This leaves behind all the convective activity which is associated with shallow clouds of cumulus congests and stratocumulus type. Tropical convective systems are known to involve a rather diverse population of cloud types and one needs to account for all of them in order to represent the life-cycle of organized convection. According to the authors, the whole argument for choosing to simple a flux-error database instead of the more or less established Stochastically Perturbed Parameterization Tendency (SPPT) is rooted from the fact that the error fluxes associated with different variables are only weakly correlated (if they are at all). However, the way they do the sampling while it does assume such correlation it makes it systematic since they sample the grid columns and not the different fluxes independently as illustrated in Figure 8. The paper has many other incoherent statements to the point that it is not at all clear what the authors want us to learn from their study.

For these reasons and many others (in the specific comments), I personally cannot recommend this paper for publication.

Specific comments

Lines 20-25 of page: This paragraph is misleading when first reading it the following question came to my mind: "Something is not quite right. How can one compare fluxes between two different models that do not necessarily go through the same integral curve in the state space?" It is only after I got to page 4 that I found out that the authors are doing the right thing by comparing flux deviation after only the first step. This needs

to be stated before hand so not to confuse the reader. Line 15, Page 4: "Therefore, the retrieved model error should rather be seen as a lower bound on the error made in the representation of the physical process." This can be interpreted as that the authors are trying to do better than the reference? This may not be possible since the direction of error can not be quantified in such a large dimensional state space! Page 4, line 27: The use of a hydrostatic model at 4km resolution needs caution–while I doubt that it can be justified, the authors are requested to provides a few words warning their readers that this is not at all realistic! This is intact a serious flaw in this study. A quick look at the Gerard et al. reference reveals indeed that the cumulus scheme on which this study is based tries to represent non hydrostatic effects (their Eqn. 5), thus it is not surprising if the deficiencies in the NPC model are have more to do with the use of hydrostatic model other than anything else. Line 25, page 5: "This database is not only useful to investigate the statistics of the model error due to deep convection parameterization (Sect. 2.3), but it will also be the basis for a stochasticperturbation scheme that can be applied in an ensemble prediction system (Sect. 3)." Rephrase or delete the whole sentence. It adds nothing to the paper it can only confuse your readers. Isn't the later statement the main objective of the study? Figure 2: This figure can be clearer. It took me maybe 5 minutes of staring at it before I could make a clear sense of it. The caption could be used to explain the labels and the color coding. Page 6, line 3: Aren't 72 evaluations too few given there is a high level of correlation in space and time because of the nature of organized convection? Figure 3: What does the label error in red stand for? It isn't clear at all. The red dots are hardly visible and they don't constitute and error but their difference does. Maybe draw a red line segment between the two red markers to indicate the error. Page 6, line 5 — page 7, line 9: The discussion in these two paragraphs and Fig 3 seems to be included in order to make the final statement that "Therefore, the total transport flux difference one time step after the switch can be considered as a representative measurement of the error in the transport flux as defined in Eq. (1)." 1) This is empirical observation has no scientific value as such. 2) The model error as defined in (1) is only valid when evaluated at first

step because the states of the two simulations change in subsequent steps. Page 8, line 5: This is not a surprise because the model needs to conserve the water budget. Page 11, line 5: This is not a surprise at all because the model needs to conserve the water budget. Page 13, lines 13-14: The way the sampling is done is not at all clear. 1. Figure 5 has three distributions, which one is actually sampled. 2. Figure 8, has six fluxes how the two are reconciled? Are you sampled the distributions in Fig.5 or the "grid columns data base"? If it is the later how are you doing it? Is it uniformly over all grid columns? Also Conditioning on the basic state would been more appropriate if one wants to genuinely emulate the cumulus scheme. Nonetheless the "success" of the completely random sample in reproducing the results implies that the cumulus parameterization is perhaps not sensitive enough to the environment, which may be problematic. Page 14, lines 1-2: Why are you doing this? Aren't the cases with zero or weak updraft part of the physics of the problem? This is clearly biased and it is not at all justified. It undermines the role of shallow cumulus and cumulus congests clouds since your distributions in Fig. 5 are based on 250hPa errors. Page 14, lines 8-21: So you are using a convection trigger. Are the two criteria enforced simultaneously or are you using one at a time? Why these particular choices? How do they compare to what the original cumulus scheme does? Figure 9, caption: "Lead times where the ensemble mean RMSE is significantly lower than the NCP control RMSE at the 95 % confidence level are indicated with a filled circle." This is not clear that this is actually true. Maybe showing the absolute errors instead would be more clearer. In any case the difference between the compared errors is probably very small. What is the actual gain really is? Page 15, line 5: This may have something to do with perhaps the fact that you are sampling the flux errors at 250 hPa in Fig. 5. Page 15, lines 7&9: CP—>NCP Page 15, lines 12-13: When and where the error in the reference configuration was it defined? You can't really tell since you are not comparing to anything else but the CP run. Please delete this sentence. Page 17, lines 4-7: This is in contradiction with the claim made upfront that a stochastic parameterization would increase the spread by accounting for model error. Page 17, lines 11-13: This applies to any ad hoc and nonphysically based CP. Page 18, line 15: Where are you looking? Do you mean MOCON and OMEGA? Page 22, lines 5-6: What does this mean? Are the two models evaluated and compared elsewhere? If so please provide the reference and eventually say for which case study it was done. It makes a huge difference if that was done for tropical or non tropical convection site. Otherwise simply delete this sentence. It simply says that in the gray zone the role of a CP is unclear whether it is beneficial or detrimental and this is already known for many years. Page 22, line 7: This isn't true. Figure 16 actually shows the opposite. The NCP ensemble is better than the MOCON ensemble during the first 9 hours. Page 24, lines 4-5: " but for many variables it even outperforms the ensemble system with the deep convection scheme switched on." Where is this shown? Page 24, line 10: spell out EPS

---

## Author Comment (AC2) · 9 Oct 2019

**General reply to reviewers 1 and 2**

The comments of the reviewers point out three main weaknesses of our manuscript:

- There is a lack of the description of the theoretical basis of the work. This was already addressed in our previously published Authors Comments 1 (AC1).

- The description of the sampling method is not sufficiently clear to the reviewers.

- The manuscript lacks a justification for the use of the model in a hydrostatic configuration.

All of these issues can be addressed. The first two are a matter of redaction. For the third, we will perform non-hydrostatic runs. Since the sampling method is very computationally demanding, this will take some time and we can only provide a new version of the manuscript after these have been performed.

In order to clarify some misconceptions in the comments of the reviewers, we provide some detailed replies below. We also describe the issue of the use of the hydrostatic dynamics and the way how we will address it.

**Theoretical basis**

comments from reviewer 1:
"The difference between the two model configurations is not the model error, but rather one representation of model uncertainty. At 4 km resolution, this uncertainty will pertain systematic differences between the two configurations chosen in this study, and sampling from the data base would mean consistently sample perturbations with the same systematic error. Using the differences between two configurations where one has a known systematic deficiency needs to be better justified, if at all possible."

and

"Another aspect that is rather confusing in the experiment design is why the study is constructed such that the perturbations are applied to the model configuration that has no deep convection at 4 km, when the operational model uses the convective parameterization? Wouldn't it be more desirable to create a perturbation scheme that could be applied to the operational ensemble system at that resolution?"

and

"The main short-comings of this study is the computation of the "error" (uncertainty) itself. Here the authors turn off the deep convection parameterization and claim "it

is assumed that the turbulence (together with shallow convection) and resolved condensation schemes might compensate for the absence of parameterized convective transport". And they proceed to compute the "error" as the difference in total transport (where one experiment is now missing the convective transport terms). This assumption is highly questionable. Just because there is no parameterized convection contributing to the transport flux of e.g. specific humidity, doesn't mean that there is no convective transport. In the "no parameterized convection" experiment this is now taken care of by the resolved dynamics, and the "compensation" discussed will be seen in the tendency of the dynamics. In fact, the authors do point out in the introduction that studies have shown that turning off the convective parameterization at âĹij4 km can lead to unrealistically strong updrafts. What is the scientific justification for systematically adding a positive (or stronger negative) perturbation to the total transport when the convective transport is missing by construction, and is now resolved?"

from reviewer 1.

**Reply**:

This confusion originates from the fact that the theoretical basis has not been well described in the manuscript. We have already provided an online reply about this to reviewer 1 in AC1.

While it is true that in practice model-error perturbations should be added the the operational model (here with parameterization), the choice for using our *academic* setup, where the model error is artificially introduced by comparing a model-without-parameterization (target) with a model-with-parameterization (truth), has the advantage that the source of the model uncertainty can be singled out and its effect on the simulations studied in detail (see also our reply in AC1). We believe that this is an interesting example of how a specific process-related model error could look like and how it could be modeled. The academic nature of the work was the reason we chose

to submit the work to NGP as opposed to more forecast-related journals as MWR and WAF.

We emphasize that we are looking for the sources of the model error. When a model-with-convective-parameterization and a model-without-convective-parameterization both perform one time step in the dynamics (starting from identical initial conditions) their results will be identical. It is only after applying the parameterization schemes that the results will differ. Considering the model-with-parameterization as the truth, the absence of a convective flux is thus by definition the (source of the) model error. In the model-without-parametrization, the absence of the stabilizing convective fluxes will, of course, trigger a response in the dynamics as the instabilities grow large enough to be resolved, creating the typically too strong updraft. Correcting for this model error at the source, by adding the stronger negative perturbations, will thus stabilize the air, preventing the growth of the instabilities leading to the too large updrafts.

**The description of the sampling method**

comments: "While I am likely confused by their narrative, the choose of the 250hP level as a reference for "sampling the grid column database" is not only not justified by the authors but it is also not accurate. This leaves behind all the convective activity which is associated with shallow clouds of cumulus congests and stratocumulus type. Tropical convective systems are known to involve a rather diverse population of cloud types and one needs to account for all of them in order to represent the life-cycle of organized convection."

and

"The way the sampling is done is not at all clear. 1. Figure 5 has three distributions, which one is actually sampled. 2. Figure 8, has six fluxes how the two are reconciled? Are you sampled the distributions in Fig.5 or the "grid columns data base"?"

from reviewer 2.

**Reply:**
The database consists of grid-columns, these entire grid-columns are sampled based on moisture convergence or CAPE (both criteria are used separately). While moisture convergence and CAPE are both typical indicators for convective activity, their use as a sampling criterion is indeed introduced quite arbitrarily. The grid-columns from the database are sampled uniformly and the error fluxes will therefore by definition represent the distributions contained in the database.
The distributions on the 250 hPa level are, however, just an example of what the distributions look like and are not used for the sampling. We understand this might confuse the readers and will clarify this better in a future version.

comment:

"According to the authors, the whole argument for choosing to simple a flux-error database instead of the more or less established Stochastically Per- turbed Parameterization Tendency (SPPT) is rooted from the fact that the error fluxes associated with different variables are only weakly correlated (if they are at all). How- ever, the way they do the sampling while it does assume such correlation it makes it systematic since they sample the grid columns and not the different fluxes indepen- dently as illustrated in Figure 8."

from reviewer 2

**Reply:**
Concerning the comparison to the multiplicative perturbations used in SPPT, we only make the claim that, looking at the correlation between the flux errors and the corresponding fluxes themselves, there is no support to use multiplicative perturbations as is done in SPPT. We do not make any statements about the SPPT method itself nor do we use this as an argument for our method of using a simple flux-error database. Furthermore, this claim is not in contradiction to sampling grid-columns (keeping the inter-flux correlations), as there is a strong correlation between the different flux errors

(see table 1 in the manuscript).

comment:

"Page 14, lines 1-2: Why are you doing this? Aren't the cases with zero or weak updraft part of the physics of the problem? This is clearly biased and it is not at all justified."

**Reply:**
The idea behind doing this, is to remove the grid-columns with no convective activity (a vertical updraught flux of 0.5 Pa/s is much smaller than the typical updraught flux in columns with convective activity), since at these columns there is also no error present, to reduce the database to a manageable size. (Keep in mind that shallow convection is handled by the turbulence scheme in the ALARO model and is thus correctly represented in all configurations).

**The use of the non-hydrostatic dynamical core**

comments:
"The simulations should be made with non-hydrostatic dynamics, for the dynamics to be able to (have a chance) to realistically simulate vertical motions generated by convection."

from reviewer 1 and comment:

"The use of a hydrostatic model at 4km resolution needs caution–while I doubt that it can be justified, the authors are requested to provides a few words warning their readers that this is not at all realistic!"

from reviewer 2.

**Reply:**
The ALADIN System can be run both with a hydrostatic and a non-hydrostatic dynamical core (Termonia et al 2018). We have performed simulations for the setup and the test period of the paper both with the hydrostatic and the non-hydrostatic version and

found that the difference are very small, see Figs. 1 and 2 at the bottom of this document where the RMSE of the three configurations (hydrostatic-with-deep-convection-parameterization (CP), hydrostatic-without-deep-convection-parameterization (NCP) and non-hydrostatic-without-deep-convection-parameterization(NH-NCP) ) is compared for different variables (filled circles indicate significant differences w.r.t. the NCP configuration)

As the reviewers know very well running EPS systems is very computationally demanding. We have chosen the hydrostatic version since it is more stable and allows thus longer time steps to perform this research being convinced ourselves that the results would not substantially be different than for the non-hydrostatic setup.

We plan to apply the found stochastic method in the non-hydrostatic version of the model. While we are convinced that the results will be essentially the same, we will do so to remove any doubts that could be raised concerning this issue. This will take time. We will write a new manuscript where the final results will be presented for the non-hydrostatic setup.

**specific comments**

The specific comments will be addressed in a new manuscript after the non-hysdrostatic runs have been performed.

[Figure]

**Fig. 1.** 500 hPa RMSE of the different model output variables for the hydrostatic configurations with (green) and without (blue) convective parameterization and the non-hydrostatic configuration (red),

[Figure]

**Fig. 2.** 850 hPa RMSE of the different model output variables for the hydrostatic configurations with (green) and without (blue) convective parameterization and the non-hydrostatic configuration (red),

---

## Editor Comment (EC1) · Juan Restrepo (Editor) · 4 Feb 2020

I have carefully re-read the expert's reviews and the Author's recent resubmittal. The referees were critical of the results in the original submittal. However, the paper has been reframed in the re-submittal with a significant positive outcome. The new submission also has new results, which potentially improves the significance of the paper. Prior to accepting, however, I am going to request that the referees take a look at the re-framing and the new results before making a determination on acceptance or for requesting further changes. Best Wishes,

Juan Restrepo

---

## Author Response (AR1)

-The difference between the two model configurations is not the model error, but rather one representation of model uncertainty. At 4 km resolution, this uncertainty will pertain systematic differences between the two configurations chosen in this study, and sampling from the data base would mean consistently sample perturbations with the same systematic error. Using the differences between two configurations where one has a known systematic deficiency needs to be better justified, if at all possible.

*We now explain what we mean by model error in a completely revised version of section 2.1.*

-The main short-comings of this study is the computation of the "error" (uncertainty) itself. Here the authors turn off the deep convection parameterization and claim "it is assumed that the turbulence (together with shallow convection) and resolved condensation schemes might compensate for the absence of parameterized convective transport". And they proceed to compute the "error" as the difference in total transport (where one experiment is now missing the convective transport terms). This assumption is highly questionable. Just because there is no parameterized convection contributing to the transport flux of e.g. specific humidity, doesn't mean that there is no convective transport. In the "no parameterized convection" experiment this is now taken care of by the resolved dynamics, and the "compensation" discussed will be seen in the tendency of the dynamics. In fact, the authors do point out in the introduction that studies have shown that turning off the convective parameterization at ∼4 km can lead to unrealistically strong updrafts. What is the scientific justification for systematically adding a positive (or stronger negative) perturbation to the total transport when the convective transport is missing by construction, and is now resolved?

*We have completely revised section 2.2. In fact when switching off the deep convection (DC) scheme, the dynamics starts to handle some of the convective transport that would otherwise be treated by the DC scheme. As is now better explained we study the error that is created in the transport when switching off the DC scheme.*

*Some models run without a DC scheme at resolutions below 5-km. We can also run our model at 4-km resolution without DC scheme. We then demonstrate that an ensemble approach can increase forecast skill.*

-The simulations should be made with non-hydrostatic dynamics, for the dynamics to be able to (have a chance) to realistically simulate vertical motions generated by convection.

*We have taken the necessary time to run the experiments with the non-hydrostatic version (NH) of the model and show the results now in comparison to the hydrostatic ones. First we show explicitly that our hydrostatic model performs better than the NH version for the setup of our experiments (as we expected since the model with the DC scheme was tuned for the hydrostatic setup). It is explained in the new manuscript that DC scheme compensated for some of the vertical transport in the hydrostatic version with respect to the NH one. We perturbed the NH NCP version with the database and show the results sec. 3.2. of this non-hydrostatic version. It essentially leads to the same conclusions. We invite the reviewers to check this with the results of the hydrostatic setup in the first version of the manuscript.*

-The perturbations are applied to the model considered the "target" forecast – which does not use a convective parameterization. Now you systematically introduce a larger

parameterized convective transport in a run with resolved convection. This seem to imply that the scale awareness of the model impose a reduction of the resolved convective transport, such that the improvement that you see relative to the control run (e.g. Figure 9), basically comes from again implicitly "activating" the convective parameterization (by systematically introducing a larger convective transport in the physics parameterizations).

*In fact, when switching off the DC scheme the dynamics takes over some of the transport. This is now discussed in the revised version when discussing Fig. 3.*

-Why the first time-step after turning off the deep convection parameterization is a representative time of the model uncertainty needs to be better justified. The uncertainty due to convection ought to grow as a function of lead time. Figure 3 simply shows total transport with and without convection.

*Indeed we agree that the first version of manuscript was lacking the detail of the theoretical justification and a clear definition of the model error we are studying. We have rewritten sec. 2.1. As a short answer here, we define model errors only after 1 time step, so we do not consider non-linear growth.*

-Another aspect that is rather confusing in the experiment design is why the study is constructed such that the perturbations are applied to the model configuration that has no deep convection at 4 km, when the operational model uses the convective parameterization? Wouldn't it be more desirable to create a perturbation scheme that could be applied to the operational ensemble system at that resolution?

*The underlying question is whether, in an ensemble context, one could represent the subgrid uncertainties related to a parameterization by a stochastic process. When considering the statistics of the model error in the Figures of the PDFs, one can notice some systematic dependencies, from which one could hope to characterize them systematically and to develop a more fundamental stochastic scheme that does not depend on a sampling of a database. This could be tested with some fitting of the model errors in our database. But alternatively we believe such a model-error database could be useful to feed a machine learning algorithm to discover systematic model-errors in the physics parameterization and then use them to perturb the models in an ensemble context.*

-Lastly, the perturbations in the distribution are not applicable to any general model system, but tied to this very particular experiment setup, and thus does not provide a general guidance for development of stochastic parameterizations. What happens if the model is used at 10 km or 1 km? Which configuration is now considered the 'perfect' model?

*We now describe the definition of the model error better in sec. 2.1. We explain better what is meant by "perfect model". It should now be clear that it can be applied generally to any resolution, but the database should then be recomputed. Model errors are certainly resolution dependent.*

Reviewer 2

**General comments**

The use of a hydrostatic model at 4 km and expected to represent convective flows in the tropics
realistically is, forgive the word, an aberration. The small improved seen with the use
of the stochastic are arguably due to this shortcoming more than anything else. It is
well known that although too coarse to represent the details of individual clouds, CRM
at few kilometre resolution (up to 10 km in some cases) represent well organized con-
vection in the tropics; the Japanese did their first global CRM simulation at 7 km and
got very realistic MJO, CCWs and MCSs. On the other hand it is also well known that
the hydrostatic balances messes up gravity waves at scales of 50 km and less and a
fortiori convective flows in this range.

*As mentioned above, we have provided outputs for a non-hydrostatic setup in the revised version of
the manuscript.*

The way the sampling of the flux errors is done is not very clear. While I am likely confused by
their narrative, the choose of the 250hP level as a reference for "sampling the grid column database"
is not only not justified by the authors but it is also not accurate. This leaves behind all the
convective activity which is associated with shallow clouds of cumulus congests and stratocumulus
type.

*Indeed you are right, the methodology of both the error computations and the sampling was not
clear in the first version of the manuscript. We have rewritten these section 2.1 and 2.2. providing
much more detail that was lacking. As a short reply, we do not sample at 250 hPa but we sample
entire vertical profiles. This is now made clear in the revised version.*

Tropical convective systems are known to involve a rather diverse population of cloud
types and one needs to account for all of them in order to represent the life-cycle of
organized convection. According to the authors, the whole argument for choosing to
simple a flux-error database instead of the more or less established Stochastically Per-
turbed Parameterization Tendency (SPPT) is rooted from the fact that the error fluxes
associated with different variables are only weakly correlated (if they are at all).

*No, the errors profiles are weakly correlated to the model profiles of the total transport. But you
have a good point: the text could be written more clearly. We have adapted it. We now write:
"Large correlation coefficients between the model transport flux and its error would suggest a
linear relationship ..."*

However, the way they do the sampling while it does assume such correlation it makes it
systematic since they sample the grid columns and not the different fluxes independently as
illustrated in Figure 8.

*Our excuses but we do not understand the last sentence.*

**Specific comments**

Lines 20-25 of page: This paragraph is misleading when first reading it the following
question came to my mind: "Something is not quite right. How can one compare fluxes
between two different models that do not necessarily go through the same integral

curve in the state space?" It is only after I got to page 4 that I found out that the authors are doing the right thing by comparing flux deviation after only the first step. This needs to be stated before hand so not to confuse the reader.

*Very correct, sec. 2.1 and 2.2 were badly written. As mentioned above we have reworked them thoroughly.*

Line 15, Page 4: "Therefore, the retrieved model error should rather be seen as a lower bound on the error made in the representation of the physical process." This can be interpreted as that the authors are trying to do better than the reference? This may not be possible since the direction of error can not be quantified in such a large dimensional state space!

*Also in this case, this part has been completely rewritten.*

Page 4, line 27: The use of a hydrostatic model at 4km resolution needs caution–while I doubt that it can be justified, the authors are requested to provides a few words warning their readers that this is not at all realistic! This is intact a serious flaw in this study. A quick look at the Gerard et al. reference reveals indeed that the cumulus scheme on which this study is based tries to represent non hydrostatic effects (their Eqn. 5), thus it is not surprising if the deficiencies in the NPC model are have more to do with the use of hydrostatic model other than anything else.

*As is now explained in the revised manuscript, we can also run the model with a non-hydrostatic dynamical core (NH). We have rerun our tests wit the NH dynamical core. We present scores in sec. 2.1 comparing both of them. We have implemented the perturbation in both hydrostatic and non-hydrostatic version. In the revised manuscript we provide the results with the non-hydrostatic version. Which lead, by the way, to the same conclusions as the ones for the hydrostatic model (which can be verified with respect the previous version of our manuscript).*

Line 25, page 5: "This database is not only useful to investigate the statistics of the model error due to deep convection parameterization (Sect. 2.3), but it will also be the basis for a stochastic perturbation scheme that can be applied in an ensemble prediction system (Sect. 3)." Rephrase or delete the whole sentence. It adds nothing to the paper it can only confuse your readers. Isn't the later statement the main objective of the study?

*Indeed, it is now deleted.*

Figure 2: This figure can be clearer. It took me maybe 5 minutes of staring at it before I could make a clear sense of it. The caption could be used to explain the labels and the color coding.

*We now explain the figure in great detail in the full text in sec. 2.2 to make the whole method more clear.*

Page 6, line 3: Aren't 72 evaluations too few given there is a high level of correlation in space and time because of the nature of organized convection?

*We found an improvement with this. Of course, with more evaluations we might expect even more improvements. This is a matter of computational resources. As mentioned to review 1, we plan to proceed investigating our model-error database. This study can be seen as a first feasibility/sanity test.*

Figure 3: What does the label error in red stand for? It isn't clear at all. The red dots are hardly visible and they don't constitute and error but their difference does. Maybe draw a red line segment between the two red markers to indicate the error.

*It is the first time step after switching off the deep-convection scheme. This is now better explained in the revised version of the manuscript. We think the confusion comes from the lack of description in sec. 2.1 and 2.2 of the previous version.*

At- page 7, line 9: The discussion in these two paragraphs and Fig 3 seems to be included in order to make the final statement that "Therefore, the total transport flux difference one time step after the switch can be considered as a representative measurement of the error in the transport flux as defined in Eq. (1)." 1) This is empirical observation has no scientific value as such. 2) The model error as defined in (1) is only valid when evaluated at first step because the states of the two simulations change in subsequent steps.

*Again, we define and compute the model error after 1 time step, see the new description in sec. 2.1 and 2.2.*

Page 8, line 5: This is not a surprise because the model needs to conserve the water budget.

*Indeed, this is a sanity check. But we prefer to not include this in the new manuscript.*

Page 11, line 5: This is not a surprise at all because the model needs to conserve the water budget.

*Idem as your remark above.*

In reply to your three points:
- Page 13, lines 13-14: The way the sampling is done is not at all clear.
- Figure 5 has three distributions, which one is actually sampled.
- Figure 8, has six fluxes how the two are reconciled? Are you sampled the distributions in Fig.5 or the "grid columns data base"? If it is the later how are you doing it? Is it uniformly over all grid columns? Also Conditioning on the basic state would been more appropriate if one wants to genuinely emulate the cumulus scheme. Nonetheless the "success" of the completely random sample in reproducing the results implies that the cumulus parameterization is perhaps not sensitive enough to the environment, which may be problematic.

*We have rewritten the paragraph starting with "The vertical and inter-variable correlation are preserved by organizing the flux-error profiles per grid column in the ...". In fact the perturbations are multivariate as we now explain in this paragraph.*

Page 14, lines 1-2: Why are you doing this? Aren't the cases with zero or weak updraft part of the physics of the problem? This is clearly biased and it is not at all justified. It undermines the role of shallow cumulus and cumulus congests clouds since your distributions in Fig. 5 are based on 250hPa errors.

*Since we only study model errors originating from the deep-convection scheme we will exclude these point from the model-error database. We write this explicitly in the paper. Also we think your confusion comes from a poor description of the definition of the model error in sec 2.1. of the previous version of the manuscript.*

Page 14, lines 8-21: So you are using a convection trigger. Are the two criteria enforced simultaneously or are you using one at a time? Why these particular choices? How do they compare to what the original cumulus scheme does?

*No it is not a trigger. They are two criteria to test whether the gridpoint has deep convection or not and to see whether it goes into the model-error database. This should now be clear with the sentence mention in reply to your point above.*

Figure 9, caption: "Lead times where the ensemble mean RMSE is significantly lower than the NCP control RMSE at the 95 % confidence level are indicated with a filled circle." This is not clear that this is actually true. Maybe showing the absolute errors instead would be more clearer. In any case the difference between the compared errors is probably very small. What is the actual gain really is?

*It is difference in error. So negative values mean improvements.*

Page 15, line 5: This may have something to do with perhaps the fact that you are sampling the flux errors at 250 hPa in Fig. 5.

*As mentioned above we are sampling vertical profiles not errors at 250 hPa.*

Page 15, lines 7&9: CP—>NCP

*In fact this is not shown. But it is true, the stochastic scheme beats the parameterization in the hydrostatic case. We have taken this sentence out of the manuscript since we are now showing the non-hydrostatic results.*

Page 15, lines 12-13: When and where the error in the reference configuration was it defined? You can't really tell since you are not comparing to anything else but the CP run. Please delete this sentence.

*It is now defined in sec. 2.1.*

Page 17, lines 4-7: This is in contradiction with the claim made upfront that a stochastic parameterization would increase the spread by accounting for model error.

*But it increases the spread. We only compare it here to the spread coming from the IC/LBC perturbations.*

Page 17, lines 11-13: This applies to any ad hoc and non-physically based CP.

*Indeed. But it is also the case here.*

Page 18, line 15: Where are you looking? Do you mean MOCON and OMEGA?

*Indeed, the text is wrong here. We now write: "Considering the spread in Fig. (15), for horizontal wind, ..." It confirms what ones expects. Models without deep-convection parameterization tend to produce grid point storms, and this may wrongly generated extra spread in an ensemble.*

Page 22, lines 5-6: What does this mean? Are the two models evaluated and compared elsewhere? If so please provide the reference and eventually say for which case study it was done. It makes a huge difference if that was done for tropical or non tropical convection site. Otherwise simply

delete this sentence. It simply says that in the gray zone the role of a CP is unclear whether it is beneficial or detrimental and this is already known for many years.

*Yes we now compare, CP to NCP in both hydrostatic and non-hydrostatic setups in sec. 2.1.You are right it is not clear whether it is beneficial or not. We deleted the sentence.*

Page 22, line 7: This isn't true.

*We are severe here for ourselves. We refer here to the confidence interval, the key word is "significant" in the sentence. We adapted the sentence. We now write: "There is a neutral to positive impact in skill for the MOCON ensemble (albeit inside the significance confidence intervals)."*

Figure 16 actually shows the opposite. The NCP ensemble is better than the MOCON ensemble during the first 9 hours.

*Indeed, we present it as it is.*

Page 24, lines 4-5: " but for many variables it even outperforms the ensemble system with the deep convection scheme switched on." Where is this shown?

*It is not shown. It is nevertheless true. It was a sentence from a previous draft of the manuscript.*

*Given the new figure in Fig. 1 and the detailed description in sec. 2.1. This is the take-home message of the paper. One can characterize model error due to shortcoming of the parameterization and then use it to perturb the model in an EPS sense to find that it can restore some predictability in a probabilistic sense.*

Page 24, line 10: spell out EPS

*This is done.*